# RouterInterp: Understanding Superposed Specialisation in Mixture of Experts Routing

Ilya Lasy [1]   Nora Yinuo Cai [2]   Kola Ayonrinde [3]

## Abstract

Sparse Mixture of Experts (MoE) models scale more efficiently than dense models by routing tokens to modular expert networks that are only active when relevant to the task. A leading hypothesis for the performance of MoE models is that each expert specialises in a single, coherent domain. However, interpretability efforts that assume this hypothesis have generally been unsuccessful. We propose and present evidence for an alternative account that we call the *Superposed Specialisation Hypothesis* (SSH): experts specialise in a disjoint union of fine-grained features rather than one broad domain. Leveraging the SSH, we introduce *RouterInterp*, a method for interpreting expert routing that identifies Sparse Autoencoder features most predictive of routing decisions and produces unified natural language explanations. On gpt-oss-20b, RouterInterp explains expert routing with 57% higher detection accuracy than prior token statistics based methods. This work provides a scalable method for generating concise and more accurate explanations of expert routing and increases our understanding of a previously uninterpretable component of foundation models.

## 1. Introduction

Sparse Mixture-of-Experts (MoE) transformers have emerged as a promising approach for scaling frontier language models (Cai et al., 2025; Fedus et al., 2022b; Du et al., 2022). In dense (i.e. non-MoE) transformers (Vaswani et al., 2017), each input is processed by all the parameters. In contrast, MoE models contain multiple expert networks and a routing mechanism that selects a subset of these "experts"

for each input token at each layer. (Shazeer et al., 2017) and (Liu et al., 2024) show that sparse MoE models can activate only 2–15% of parameters per input, enabling dramatic parameter scaling with minimal increases in inference cost and latency, leading to strong performance on a variety of tasks (Fedus et al., 2022a).

The strong performance of Sparse MoE models has often been attributed to expert specialisation (Lewis et al., 2021): if each expert learns to handle a subset of the input data distribution or perform only a subset of computations, then using only a subset of the parameters for each input can be both *effective* and *efficient*[1]. We call this explanation for the success of MoEs the **Specialisation Hypothesis**.

> **The Specialisation Hypothesis**
>
> Sparse Mixture-of-Experts models work well because different experts specialise in different parts of the input data distribution or different types of computation, allowing each input to be processed by only a small subset of the model's parameters.

One seemingly natural corollary of the Specialisation Hypothesis is that if each expert specialises in a particular domain then routing decisions should be human-interpretable (at least insofar as humans understand the relevant domains). For example, perhaps one expert specialises in processing medical text and contributes medical knowledge and reasoning when the input is related to healthcare. Another expert might specialise in generating mathematics and yet another in storytelling or translating romance languages (Ayonrinde, 2023a). However, prior work analysing MoE models has struggled to recover clear, interpretable patterns of expert specialisation (Jiang et al., 2024; Lewis et al., 2021; Zoph et al., 2022). We believe that this difficulty derives from not distinguishing between two distinct forms of the Specialisation Hypothesis which we term the **Domain Specialisation Hypothesis** (DSH) and the **Superposed Specialisation Hy-**

---

[1]Faculty of Informatics, TU Wien, Vienna, Austria [2]Independent [3]UK AI Security Institute, London, UK. Correspondence to: Ilya Lasy <ilya.lasy@tuwien.ac.at>.

*Proceedings of the 43rd International Conference on Machine Learning*, Seoul, South Korea. PMLR 306, 2026. Copyright 2026 by the author(s).

---

[1]*Effective* because all the processing a token requires is handled by the selected experts, and *efficient* as only a small fraction of the total parameters and associated FLOPs are used for each forward pass.

**pothesis** (SSH), respectively. The distinction between these hypotheses becomes apparent when we consider that real-world data contains far more fine-grained categories (e.g., specific topics, syntactic constructions, reasoning patterns) than any MoE layer has experts [2]. This means multiple such categories, which we call *micro-domains* [3], must inevitably be routed to the same expert [4]. Under the *Domain Specialisation Hypothesis*, semantically similar micro-domains cluster within an expert, yielding a coherent domain that the expert specialises in[5]. In contrast, under the *Superposed Specialisation Hypothesis*, an expert specialises in a disjoint collection of features spanning multiple unrelated micro-domains. This mirrors the *superposition* phenomenon observed in neural networks, where multiple features are represented in the same set of neurons (Elhage et al., 2022).

> **Alternative Specialisation Hypotheses**
>
> **Domain Specialisation Hypothesis:** Sparse Mixture-of-Experts models work well because different experts specialise in different semantically coherent domains (e.g. medical text, mathematical reasoning, romance language translation).
>
> **Superposed Specialisation Hypothesis:** Sparse Mixture-of-Experts models work well because different experts specialise in disjoint collections of features corresponding to multiple micro-domains that are not highly semantically related.

To test these hypotheses, we need a way to identify the fine-grained features that drive each expert's activation. We use sparse autoencoders (SAEs) (Makhzani & Frey, 2014; Cunningham et al., 2024; Bricken et al., 2023) to decompose model activations into sparse latent representations (SAE latents), where each latent corresponds to an interpretable feature. These SAE latents serve as a feature-level proxy for the micro-domains in our hypotheses. SAE latents predict which experts are selected far more accurately than token co-occurrence statistics or sparse linear probes on alternative bases of the residual stream. SAE-based classifiers achieve 0.74 and 0.73 macro-F1 in routing prediction on OLMoE-1B-7B and gpt-oss-20b, respectively.

Treating each expert's routing-predictive latents as its micro-domains (Section 3), we find evidence for the Superposed Specialisation Hypothesis (SSH): our experiments show that expert routing can only be explained in terms of multiple semantically disjoint micro-domains, not a single monosemantic domain (Section 4.3). We also present theoretical arguments for the SSH (Section 4).

Based on these findings, we develop **RouterInterp**, which interprets routing as arising from a *combination* of SAE features: it selects the latents most predictive of each expert's routing, generates concise natural-language explanations for each, and aggregates them into an expert-level explanation (Figure 1). To evaluate whether these explanations are human-understandable, we measure how accurately a language model can predict expert activation given only the explanation (Section 5.1). On gpt-oss-20b, RouterInterp attains a 0.61 explanation score, outperforming explanations based on token-expert co-occurrence (0.40). By interpreting routing as a superposition of interpretable SAE features rather than a single domain, RouterInterp demonstrates that we can meaningfully understand MoE routing decisions even when experts specialise in disjoint collections of features.

Our contributions are as follows:

- We show that SAE latents predict MoE expert routing with 0.74 and 0.73 macro-F1 on OLMoE-1B-7B and gpt-oss-20b, substantially better than token-level statistics (Section 3).

- We introduce the *Superposed Specialisation Hypothesis (SSH)*: the idea that MoE experts specialise in disjoint, semantically heterogeneous collections of features rather than a single coherent domain. We also give two theoretical arguments for why this holds in practice (Section 4).

- We provide empirical evidence for the SSH: for each expert, the SAE latents that best predict its activation span semantically distinct clusters, not a single coherent domain (Section 4.3).

- We present *RouterInterp*, our SAE-based method, which produces accurate and concise explanations of routing decisions, achieving a 0.61 explanation score on gpt-oss-20b (Section 5).

## 2. Background

### 2.1. Sparse Autoencoders (SAE)

A fundamental challenge in interpreting LLM activations is *superposition*: models encode more features than available dimensions, resulting in *polysemantic neurons* that activate for multiple unrelated concepts (Elhage et al., 2022). Sparse Autoencoders (SAEs) address this by mapping an activation

---

[2]Typical MoE models use between 8 and 128 experts per layer (Jiang et al., 2024; Lewis et al., 2021; Zoph et al., 2022).

[3]For example, "medical text" may be a broad domain composed of micro-domains such as "anatomy", "diseases", and "medical procedures".

[4]by the Pigeonhole Principle (Rebman, 1979)

[5]We may think of a domain as expressible as a cluster of similar micro-domains.

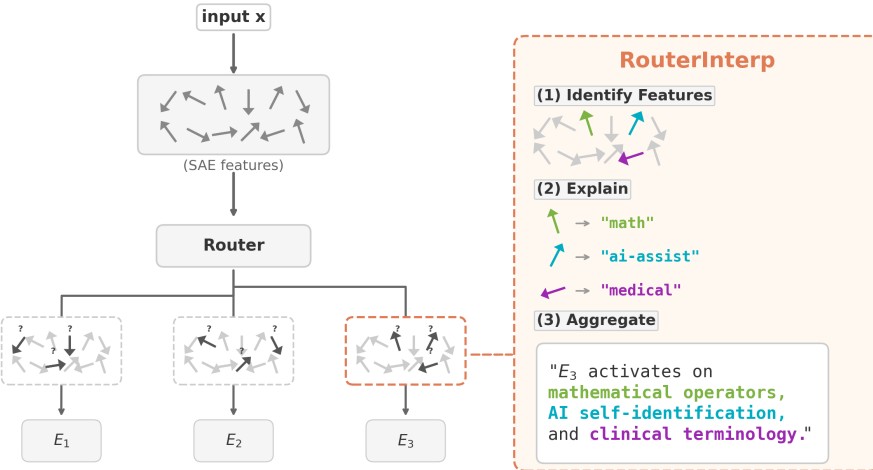

*Figure 1.* RouterInterp explains routing decisions as a combination of interpretable features. Sparse autoencoder (SAE) latents, sparse feature representations extracted from model activations, pick out directions in activation space that correspond to interpretable features. RouterInterp *identifies* SAE latents that are most predictive of routing, *explains* these features in concise natural language strings, and *aggregates* these explanations into an expert-level explanation. While no single monosemantic concept captures an expert's behavior, aggregating multiple features makes routing understandable.

vector $\boldsymbol{x} \in \mathbb{R}^N$ to a sparse latent representation $\boldsymbol{z} \in \mathbb{R}^F$ (where $F > N$). Ideally, $\boldsymbol{z}$ represents a "disentangling" of compressed representations into monosemantic, single-concept latents referred to as *features* (Bricken et al., 2023; Cunningham et al., 2024).

The SAE architecture consists of an encoder that projects activations into a sparse latent space, and a decoder that reconstructs the original activation from the sparse latents:

$$\boldsymbol{z} = \sigma(\boldsymbol{W}_{\text{enc}}(\boldsymbol{x} - \boldsymbol{b}_{\text{pre}}) + \boldsymbol{b}_{\text{enc}}) \tag{1}$$

$$\hat{\boldsymbol{x}} = \boldsymbol{W}_{\text{dec}}\boldsymbol{z} + \boldsymbol{b}_{\text{pre}} \tag{2}$$

where $\boldsymbol{W}_{\text{enc}} \in \mathbb{R}^{N \times F}$, $\boldsymbol{W}_{\text{dec}} \in \mathbb{R}^{F \times N}$, and $\sigma$ is a sparsifying activation function. We follow Gao et al. (2025)'s **Top-K SAE** approach, where $\sigma = \text{TopK}(\cdot, s)$ retains only the $s$ largest activations per input and zeros out the rest, ensuring that the latent is sparse[6]. The SAE is trained to optimise reconstruction fidelity $\mathcal{L} = \|\boldsymbol{x} - \hat{\boldsymbol{x}}\|_2^2$, ensuring that the original activation can be accurately reconstructed from the sparse latent representation $\boldsymbol{z}$.

### 2.2. Mixture of Experts (MoE)

In a standard MoE layer, the dense feed-forward network (FFN) is replaced by $E$ parallel expert networks $\{\boldsymbol{E}_i\}_{i=1}^E$, each typically an FFN itself. A learned router network ($\boldsymbol{W}_r$) determines which experts process each token. Given an input token representation $\boldsymbol{x}$, the router computes routing logits $h(\boldsymbol{x}) = \boldsymbol{W}_r \cdot \boldsymbol{x}$ and converts them to a probability

distribution over experts via softmax:

$$p_i(\boldsymbol{x}) = \frac{\exp(h(\boldsymbol{x})_i)}{\sum_{j=1}^E \exp(h(\boldsymbol{x})_j)} \tag{3}$$

To enforce sparsity, only the top-$k$ experts with the highest probabilities are selected. Letting $\mathcal{T}$ denote the set of selected expert indices, the layer output is computed as the weighted sum of expert outputs:

$$\boldsymbol{y} = \sum_{i \in \mathcal{T}} p_i(\boldsymbol{x}) \cdot \boldsymbol{E}_i(\boldsymbol{x}) \tag{4}$$

A key challenge in MoE training is *load balancing*: without intervention, models tend to collapse to using only a few experts, leaving others undertrained (Shazeer et al., 2017). This is addressed through auxiliary losses that encourage uniform expert utilization.

## 3. SAE Features as Routing Predictors

**SAE-Based Routing Prediction.** We first evaluate whether SAE features can linearly predict routing patterns. We extract activations $\boldsymbol{x}$ from the residual stream before the routing function and use an SAE encoder to produce sparse feature representations $\boldsymbol{z}$. We then train a logistic regression classifier on these sparse feature representations, one per expert $E_i$, to predict whether a given expert is activated for some context. Each classifier is trained with binary cross-entropy against the ground-truth binary label indicating whether $E_i$ appears in the router's top-$k$ selection for that token[7].

---

[6]To keep this distinct from MoE routing sparsity, we reserve $s$ for SAEs and use $k$ for how many experts are activated per token (Section 2.2).

[7]We additionally show that classifiers recover the router's full probability ordering, not just top-$k$ membership in Section J.

To investigate how routing information is distributed across features, we additionally train classifiers using only the $m$ most active features per token (ranked by activation magnitude). We find that routing prediction improves consistently with more features up to $m{=}32$–$64$, which indicates that multiple concepts are routed to the same expert in superposition (Section B).

**Baselines.** We formalise and extend prior token-level analyses as an **n-gram baseline**: for each token in the vocabulary, we count how often it co-occurs with each expert's activation on a training set, and at inference predict the experts it most frequently co-occurred with. We extend prior unigram-only analyses to also include bigrams (current and previous token), allowing us to assess whether local token context improves prediction. To distinguish whether prediction gains stem from the learned SAE basis or simply from access to continuous residual stream representations, we additionally compare against sparse linear probes on two alternative bases. The **neuron basis probe** retains only the top-$s$ highest-activating neurons from the residual stream, setting all others to zero, and trains a linear classifier on this sparse representation. The **PCA basis probe** first projects the residual stream onto its leading principal components, then applies the same top-$s$ sparsification in this rotated basis. Here probe sparsity is required to fairly compare against the sparse SAE-based classifier.

**Feature Selection.** To identify which SAE features are most predictive of each expert's activation, we use the '$\rho$-usefulness' metric (Ilyas et al., 2019), a data-driven selection criterion that measures how discriminative a feature is for expert selection. A feature $f$ is $\rho$-useful for expert $E_i$ if the expected activation of $f$ is higher when expert $E_i$ is selected than when it is not:

$$\rho(f, E_i) = \mathbb{E}[f(\boldsymbol{x}) \mid E_i \in \mathcal{T}(\boldsymbol{x})] - \mathbb{E}[f(\boldsymbol{x}) \mid E_i \notin \mathcal{T}(\boldsymbol{x})] \tag{5}$$

where $\mathcal{T}(\boldsymbol{x})$ denotes the set of selected experts for input $\boldsymbol{x}$. Features with high $\rho$-usefulness scores are specifically predictive of a particular expert's activation. For each expert $E_i$, we select the top-$n$ features ranked by $\rho$-usefulness score, denoting this set of features $\mathbb{F}_i$. [8] We additionally run intervention experiments to confirm that high-$\rho$ features have a *causal* effect on routing: positive steering along their directions increases expert activation and negative steering suppresses it, compared to random feature sets for most experts (Section D).

|                    | OLMoE-1B-7B | gpt-oss-20b |
|--------------------|:-----------:|:-----------:|
| Unigram Baseline   | 0.564       | 0.295       |
| Bigram Baseline    | 0.633       | 0.355       |
| Neuron Basis Probe | 0.666       | 0.586       |
| PCA Basis Probe    | 0.680       | 0.536       |
| SAE Predictor (Ours) | **0.740** | **0.729**   |

*Table 1.* SAE latents predict MoE routing better than token-level statistics and sparse probes. Neuron and PCA basis probes operate on sparse residual stream representations matched in sparsity to the SAE. N-gram baselines predict routing from token co-occurrence frequencies. Results reported over $\sim$1M tokens as macro-F1 for layer 11 of OLMoE-1B-7B and layer 12 of gpt-oss-20b.

**Experimental Setup.** We evaluate on two MoE architectures: OLMoE-1B-7B (Muennighoff et al., 2025) and gpt-oss-20b (Agarwal et al., 2025). For training OLMoE SAEs, we use OLMoE-mix-0924 (Muennighoff et al., 2025). For routing prediction, we collect activations using the Pile (Gao et al., 2020)[9] dataset. For OLMoE, we train Top-K SAEs for 100M tokens on activations sampled from layers 3, 7, 11, and 15 (out of OLMoE's 16 layers), with per-token latent sparsity $s = 32$ and 32,768 features. For gpt-oss (24 layers), we use trained BatchTopK SAEs[10] (Lin, 2025) on layers 4, 8, 12, 16, and 20, with 131,072 features and sparsity $s \in \{64, 128\}$. We report SAE training metrics in section G.

Linear classifiers are trained on SAE features from 1M tokens. For gpt-oss, sparse probe baselines are matched to the SAE sparsity variants ($s = 64$ and $s = 128$). We report *macro-F1*: the mean per-expert F1 for binary top-$k$ membership.

**Routing Prediction Results.** To show that learned sparse features are capable of predicting expert routing accurately, we compare SAE-based prediction against n-gram baselines and sparse probes. Table 1 shows two findings. First, token-level statistics are a poor proxy for expert routing, with unigram and bigram baselines lagging far behind SAE latents on both OLMoE-1B-7B and gpt-oss-20b. Second, at matched sparsity the SAE predictor also outperforms both neuron-basis and PCA-basis sparse probes on each model, showing the gain is specific to the learned SAE basis rather than any linear readout of the residual stream. We show in Section E that this SAE-basis advantage extends across multiple layers and sparsity levels ($s\in\{64, 128\}$) of gpt-oss-20b, where the SAE beats the best neuron and PCA probe at four of five layers. Since all three probes use the same residual activations and differ only in the basis where sparsity is

---

[8]We evaluate the effect of feature set size $|\mathbb{F}_i|$ on explanation quality and compare against an alternative selection method based on cosine similarity between router weight vectors and SAE decoder directions in Section C.

[9]`https://huggingface.co/datasets/monology/pile-uncopyrighted`

[10]`https://huggingface.co/andyrdt/saes-gpt-oss-20b`

*Figure 2.* Illustration of two alternative hypotheses for the expert specialisation in MoE models, (a) the Domain Specialisation Hypothesis (DSH) and (b) the Superposed Specialisation Hypothesis (SSH). (a) Under the DSH, dataset examples (points) from the same semantic domain (coloured regions) are routed to the same expert (they all have the same colour). In this case experts specialise in a single coherent domain. (b) Under the SSH, however, examples routed to the same expert (i.e. have the same colour) are scattered across multiple disjoint domains (different coloured regions). Here the clusters of adjacent points that are routed to the same expert represent *micro-domains*: fine-grained features of the input dataset. In this case, experts do not specialise in a single domain but instead specialise in a disjoint union of fine-grained features (micro-domains). We suggest that the SSH is a more accurate description of expert specialisation in practice.

enforced, this supports the view that the SAE features carry routing-specific structure, motivating RouterInterp's use of SAE decomposition.

## 4. Specialisation Hypotheses

In Section 1, we introduced two ways to operationalise the idea of MoE models being effective due to specialisation in terms of the *Domain Specialisation Hypothesis (DSH)* and the *Superposed Specialisation Hypothesis (SSH)*. These hypotheses can be understood as claims about whether expert routing is *monosemantic* (can be explained by a single concept or domain) or *polysemantic* (can only be explained by a collection of multiple concepts or domains which are relevant in different contexts). The idea of superposition is inspired by Elhage et al. (2022), who describe a theory of the superposition of neurons[11]. We extend this idea to the superposition of *experts*, arguing that each expert may be specialized in a disjoint collection of domains.

In this section, we formalise the two hypotheses (DSH and SSH), state their predictions, give theoretical motivations for why SSH might hold in practice, and present experiments that distinguish the two hypotheses.

---

[11]Arora et al. (2018) and Goh (2016) both find evidence for superposition-like phenomena and utilise this structure for interpretability similarly to our work.

### 4.1. Alternative Specialisation Hypotheses

Suppose that we have a set of $D$ micro-domains occurring in a corpus $\mathcal{C}$, denoted $\mathcal{D} = \{d_1, \ldots, d_D\}$[12]. Suppose also that we have an MoE layer with $E$ experts.

Firstly, note that when $D = E$, that is there are the same number of micro-domains as experts, then we should expect the experts to specialise in a single micro-domain. This is consistent with both the DSH and SSH. Secondly, when $D < E$, that is there are fewer micro-domains than experts, then we should expect multiple experts to specialise in the same micro-domain or for there to be redundant experts which are never routed to. This is also consistent with both the DSH and SSH.

However, in the more interesting and realistic case of $D > E$, the two hypotheses make different predictions. Multiple micro-domains must be routed to the same expert and so the question is whether each expert specialises in multiple similar micro-domains (DSH) or in multiple dissimilar micro-domains (SSH).

- **Domain Specialisation Hypothesis (DSH):** Under the DSH, similar micro-domains are routed to the same expert and so an expert specialises in a semantically coherent domain consisting of multiple adjacent micro-domains. *Prediction: Expert specialisation can be*

---

[12]Here, the domains could be topics, genres, or other semantic or syntactic categories in the data or could alternatively represent sections of the input space that require the same processing for the next layer of the model.

*readily explained in terms of a single (monosemantic) concept or domain.*

- **Superposed Specialisation Hypothesis (SSH):** Under the SSH, dissimilar micro-domains are routed to the same expert and so an expert specialises in a collection of semantically disjoint domains. *Prediction: Expert specialisation can only be explained in terms of multiple (polysemantic) concepts or domains.*

## 4.2. Theoretical Motivation for the SSH

There are two core arguments for why we might expect the Superposed Specialisation Hypothesis to be a more accurate description of expert specialisation.

**Argument 1 - Interference Minimisation.** Elhage et al. (2022) argue that superposition is an effective strategy for models because two sparse features that are not frequently co-activated can relatively unambiguously share the same set of neurons. Sparsity, and in particular, low co-activation, reduces the interference cost of superposition and so allows the model to represent more features than the number of available neurons. They find that correlated features tend to be represented orthogonally because otherwise this would lead to high levels of interference noise that would make it difficult for the model to disentangle and use these features.

Analogously, we argue that the router is incentivised to assign dissimilar domains to the same expert so that the expert can perform Computation in Superposition (Hänni et al., 2024; Linsefors & Bushnaq, 2025a;b; Newgas, 2025). In other words, a single expert can have multiple disjoint transforms that it applies depending on the input domain.

**Argument 2 - Load Balancing.** In MoE training, it is common to include a load-balancing loss that encourages the expert router to distribute the incoming tokens evenly across all experts to avoid the under-utilisation of some experts (Shazeer et al., 2017; Fedus et al., 2022a). If models are additionally trained with relatively small batch sizes,[13] then this can have the unintended effect of encouraging the expert router to route dissimilar inputs to the same expert. We can see this because a single sequence is likely to contain mainly tokens that are from the same macro-domain. Hence, if the batch size is small, then the batch will not contain very many different macro-domains; however, the load-balancing loss will encourage the router to distribute tokens from these few macro-domains across all of the experts. This necessarily requires that some tokens from the same macro-domain will be routed to different experts - in opposition to the DSH,

where we would expect all tokens from the same macro-domain to be routed to the same expert.

## 4.3. Empirical Evidence for Superposed Specialisation

If the SSH is true, then we should expect that each expert can be activated by a range of disparate features and hence that it is generally not possible to explain an expert's routing behaviour in terms of a single coherent domain. We test this by treating each expert's top-$n$ $\rho$-useful SAE latents $\mathbb{F}_i$ (Section 3) as proxies for the micro-domains routed to $E_i$ (Section 4.1), and asking whether those micro-domains are mutually similar (DSH) or mutually dissimilar (SSH).

**Per-expert micro-domain similarity.** Let $\text{sim}(f_j, f_k)$ be a similarity measure between two latents, which represent two micro-domains. The hypotheses translate directly into predictions about how the latents in $\mathbb{F}_i$ behave under sim:

- **DSH:** the $n$ latents in $\mathbb{F}_i$ are mutually similar under sim.

- **SSH:** the $n$ latents in $\mathbb{F}_i$ are mutually dissimilar under sim.

To quantify the semantic diversity within $\mathbb{E}_i$, we introduce $G(E_i) \in \{1, \ldots, n\}$, the number of semantically distinct clusters that $E_i$'s top-$n$ latents fall into. We first generate a natural-language explanation for each of the $E \times n$ top-$\rho$ latents in the layer (Section 5.1). Each explanation is embedded with a sentence transformer (`all-MiniLM-L6-v2`) and assigned to a cluster by $k$-means in a the embedding space, with $k$ chosen by silhouette score. $G(E_i)$ is the number of distinct clusters occupied by expert $E_i$'s $n$ latents. The DSH and SSH make opposite predictions for $G(E_i)$: the DSH predicts $G(E_i){=}1$ (all of $E_i$'s latents fall in a single cluster, representing one coherent micro-domain), whereas the SSH predicts $G(E_i){=}n$[14] (each latent occupies its own cluster, representing a semantically distinct micro-domain).

Figure 3 shows the distribution of $G(E_i)$ across experts at the deepest tested layer of each model. The empirical $G$ values are consistent with the SSH prediction in both models: mean $G$ is 7.8 on gpt-oss-20b (layer 19) and 7.1 on OLMoE-1B-7B (layer 14), out of an upper bound of $n{=}10$, and far from the value of 1 predicted by the DSH. Nearly every routing-predictive latent within an expert lives in its own cluster rather than sharing one with another of the expert's

---

[13]OLMoE-1B-7B uses a batch size of 4M tokens over 5.133T training tokens (Muennighoff et al., 2025): each batch covers roughly $8 \times 10^{-7}$ of the training data. The batch size for gpt-oss-20b is not publicly reported.

[14]A high $G(E_i)$ could also be evidence for the routing being *inefficient* (the router fails to exploit input structure) or *inexplicable* (router decisions are not a learnable function of semantic features). We rule out the former from MoE models' strong downstream performance, and the latter from RouterInterp (Section 5) explaining routing in terms of SAE latents.

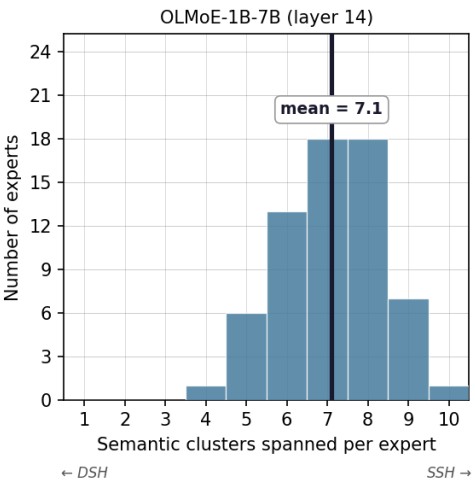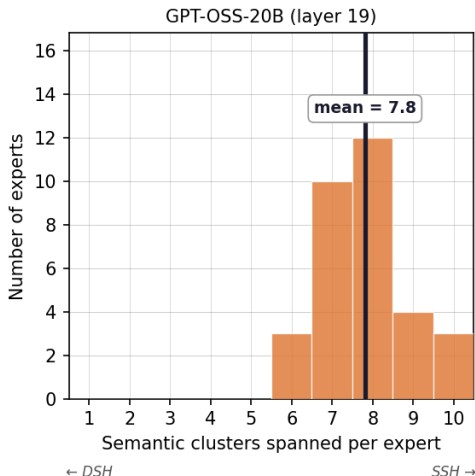

*Figure 3*. **Each expert covers a semantically diverse range of micro-domains: nearly every latent that best predict its activation occupies its own semantic cluster.** For each expert we take its top $n=10$ $\rho$-useful SAE latents $\mathbb{F}_i$, and cluster all $E \times n$ layer-wide latents by the semantic content of their natural-language explanations. We then count how many of these clusters, $G(E_i)$, each expert's $n$ latents occupy. The DSH predicts $G(E_i)=1$ (all of an expert's latents fall in a single cluster), whereas the SSH predicts $G(E_i)=n$ (each latent occupies its own cluster). The empirical values of $G$ are consistent with the SSH prediction in both models: mean $G$ is 7.8 on gpt-oss-20b (layer 19) and 7.1 on OLMoE-1B-7B (layer 14)

latents—matching the SSH prediction and inconsistent with what the DSH predicts. The same pattern holds when we measure similarity by how often latents co-fire on the same tokens rather than by the semantics of their explanations (Section F.1). Together these tests raise our prior on the SSH as the better explanation for MoE expert routing.

## 5. RouterInterp

Having established that expert routing is better described by the SSH (Section 4.3), we present **RouterInterp**, our method for generating natural language explanations of expert routing through aggregating feature explanations.

### 5.1. Method

RouterInterp operates in three stages: (1) identifying SAE features most relevant to each expert's routing behaviour, (2) generating natural language explanations for each feature, and (3) aggregating these into expert-level explanations.

**Feature Selection.** We identify which SAE features are most predictive of each expert's activation using the 'ρ-usefulness' metric (Section 3). For each expert $E_i$, we select the top-$n$ features ranked by $\rho$-usefulness score, denoting this set of features $\mathbb{F}_i$. [15].

---

[15]We evaluate the effect of feature set size $|\mathbb{F}_i|$ on explanation quality and compare against an alternative selection method based on cosine similarity between router weight vectors and SAE decoder directions in Section C.

**Feature Explanations.** Given the selected feature set $\mathbb{F}_i$ for each expert $E_i$, we generate natural language explanations for each feature following the automated interpretability framework of (Paulo et al., 2025). For each feature $f \in \mathbb{F}_i$, we present a language model (the *explainer*) with activating examples sampled using the default AutoInterp scheme: the feature's activation distribution across the dataset is divided into 10 equal-probability bins (deciles), and an equal number of examples is drawn from each bin, ensuring coverage across the full range of activation strengths. The explainer infers what concept or pattern the feature detects, producing a concise natural language description.

**Aggregation.** To obtain an expert-level explanation, we prompt a language model with all $n$ feature explanations for expert $E_i$ and ask it to synthesise a single coherent paragraph describing when the expert activates. The goal is an explanation that enables predicting when the expert activates. Prompts are provided in Section L.

**Baselines.** As an **n-gram explanation baseline**, for each expert $E_i$, we use tokens that most frequently co-occur with expert activation and prompt a language model to summarise them into a natural language description. To isolate the contribution of the SAE decomposition step, we introduce a new baseline, **Expert Activations AutoInterp**, which applies the standard AutoInterp pipeline (Paulo et al., 2025) directly to expert activations without SAE decomposition. For each expert $E_i$, we collect text samples that activate the expert and prompt a language model to identify patterns and generate a natural language explanation. This baseline

|  | Layer 4 | Layer 8 | Layer 12 | Layer 16 | Layer 20 |
|---|---|---|---|---|---|
| RouterInterp (SAE with $s$=128) | **0.59** | **0.58** | **0.59** | 0.56 | **0.61** |
| RouterInterp (SAE with $s$=64) | 0.54 | **0.58** | 0.53 | 0.53 | 0.60 |
| Expert Activations AutoInterp | 0.57 | 0.56 | 0.51 | **0.63** | 0.56 |
| Bigram AutoInterp | 0.23 | 0.47 | 0.42 | 0.35 | 0.40 |

*Table 2.* RouterInterp generates explanations of expert routing that outperform the current state of the art methods (Bigram AutoInterp) by ∼57% on average (gpt-oss-20b, macro-F1). We additionally introduce a stronger baseline, *Expert Activations AutoInterp*, which applies the same AutoInterp pipeline directly to each expert's activating contexts without any SAE decomposition. RouterInterp ($s$=128) beats it at four of five layers. Comparing our two SAE variants, the denser $s$=128 matches or exceeds $s$=64 at every layer, suggesting RouterInterp benefits from a wider SAE feature budget.

receives richer context than n-gram statistics (full text passages rather than token counts) but lacks the feature-level decomposition provided by SAEs. We report prompts used for the AutoInterp pipeline in Section L.

**Explanation Scoring.** To evaluate whether our expert explanations accurately capture routing behaviour, we adapt the AutoInterp *Detection* scoring setup from Paulo et al. (2025). In AutoInterp, another language model (the *scorer*) uses the explanation to predict expert activation on held-out examples.

We construct a balanced evaluation set for each expert $E_i$ containing *positive examples* (contexts where a token was routed to expert $E_i$) and *hard negatives* (semantically similar contexts that don't have any tokens that route to $E_i$). To find hard negatives, we embed all contexts using a sentence embedding model and rank them by cosine similarity to each positive example. We then search among the most similar contexts for those that: (1) do not contain any tokens that route to expert $E_i$, and (2) contain tokens that route to some expert $E_j$ that shares features with $E_i$ (i.e., $\mathbb{F}_i \cap \mathbb{F}_j \neq \varnothing$)[16]. For each example in the evaluation set, we present the scorer with the example's textual context and the expert explanation, asking it to predict whether expert $E_i$ would be activated. This yields a binary prediction per example, which we compare against the ground truth routing decision. We report F1 score of this binary classification task, measuring how well the synthesised explanation captures the expert's routing behaviour. We also run a small-scale human evaluation to validate if the LLM scorer is reliable: human–LLM agreement is comparable to human–human agreement, supporting its use as a proxy for human interpretability judgments (Section H).

### 5.2. Experimental Setup

We use the same models, datasets, and SAEs as in Section 3. For gpt-oss, we generate explanations across layers using the top-10 $\rho$-useful features per expert. We use the Delphi library (Paulo et al., 2025) with DeepSeek-V3.2 (DeepSeek-

AI et al., 2025) for explanation generation, summarisation, and calculating explanation scores. We use the all-MiniLM-L6-v2 model from Sentence-Transformers library (Reimers & Gurevych, 2019) for computing embeddings for samples in the evaluation set for explanation scoring.

### 5.3. Results

Table 2 shows results for gpt-oss-20b, where each expert's explanation aggregates explanations of the 10 SAE features with highest $\rho$-usefulness scores for that expert.

RouterInterp outperforms bigram-based AutoInterp at every layer and beats Expert Activations AutoInterp (explanations from activating contexts without SAE decomposition) at four of five layers with $s$=128. The one exception is layer 16, where our SAEs also exhibit the highest fraction of variance unexplained; we discuss the effect of the underlying SAE in Section G. RouterInterp's advantage is expected under the SSH: when an expert's routing spans multiple disjoint micro-domains, an LLM shown mixed activating contexts struggles to separate them, whereas explaining each feature (micro-domain) in isolation helps the explainer produce more accurate descriptions. We show example explanations in Section K.

## 6. Discussion

Our results validate the Superposed Specialisation Hypothesis (SSH) and confirm that expert routing is highly polysemantic. The set of features most aligned with any single expert often spans disjoint concepts, meaning experts do not specialize in a single monosemantic domain. This is supported by our automated expert interpretations, which frequently describe experts as activating on logical disjunctions of unrelated features [17].

A common objection to this picture is that clear, single-role expert specialisation *is* easy to find: expert pruning can

---

[16]We perform feature sharing analysis in Section I.

[17]Examples include an expert activating on both "C/C++ preprocessor directives" and "spices and herbs" or "medical conditions" and "electronic components".

drop half a model's experts with near-lossless quality on in-domain generative tasks such as code generation (Lasby et al., 2026), multilingual models contain experts that fire preferentially on particular languages (Bandarkar et al., 2026), and a small, identifiable set of experts can be toggled to reliably switch a specific behaviour, such as safety refusals, on or off (Fayyaz et al., 2026). None of this contradicts SSH. Each result shows that an expert *participates* in a language, task, or behaviour; none shows that this is the only thing the expert does. Our measurements make the gap concrete: each expert's top routing-predictive features occupy ∼7–8 disjoint semantic groups on average (Section 4.3), so any one such role is at most a small fraction of what the expert actually does.

One remaining question about how experts specialise is whether experts aggregate tokens based on *input similarity* (grouping tokens that contain similar semantic features) or *functional similarity* (grouping tokens that require similar downstream transformations). We suggest that expert specialisation based on functional similarity may be a productive way to understand routing, with each expert performing 'Computation in Superposition' (Hänni et al., 2024; Newgas, 2025). In this way, experts may group seemingly unrelated inputs that require a set of disjoint computational operations to be applied in the current layer. We would be excited about future work testing the functional routing hypothesis, as understanding the principles guiding expert specialisation could inform the design of more efficient and interpretable MoE architectures.

**Limitations.** Our method's effectiveness depends on the quality of both SAEs and the automated explanation pipeline. Concretely, RouterInterp performance correlates with SAE reconstruction quality across layers: higher FVU corresponds to weaker routing prediction and explanation scores, and gains from improving SAE sparsity are largest at the layer with the highest reconstruction error (Section G).

When aggregating the contributions of multiple features, RouterInterp produces natural language explanations that increase in size with more features. Given the length considerations for an increasing number of features, methods which further compress the explanations to be more concise might provide useful optimisations to our method. Similarly, interactive presentation formats, such as feature dashboards on Neuronpedia (Lin, 2023), may better enable exploration of the feature-expert relationships that drive routing decisions.

Finally, we note that our evaluation was performed on two models (OLMoE-1B-7B and gpt-oss-20b) in the 1B–20B parameter range, and we would be excited about future work scaling RouterInterp to frontier architectures with more parameters, shared experts and different routing methods.

## 7. Conclusion

We introduced *RouterInterp*, a method that interprets MoE routing decisions by identifying sparse autoencoder features most predictive of expert selection and aggregating their explanations into unified natural language descriptions. RouterInterp produces concise descriptions that capture not just *which* tokens route to an expert, but *why*. On gpt-oss-20b, our results confirm that SAE features provide both predictive accuracy for routing and semantic interpretability: (i) SAE-based classifiers predict routing with 0.73 macro-F1 vs. 0.59 for sparse linear probes. (ii) RouterInterp's natural language explanations achieve a 0.61 explanation score, outperforming both explanations generated directly from expert-activating text without SAE decomposition (0.57) and bigram-based explanations (0.40).

RouterInterp emerged from formalizing two alternative specialisation hypotheses: the *Domain Specialisation Hypothesis* (DSH), which posits that experts specialize in semantically coherent domains, and the *Superposed Specialisation Hypothesis* (SSH), where experts respond to disjoint collections of unrelated micro-domains. Our experiments provide evidence supporting the SSH: experts are highly polysemantic, with the set of features most aligned with any single expert often spanning disjoint concepts and/or domains. This is supported by our automated expert interpretations, which frequently describe experts as activating on logical disjunctions of unrelated features.

Several directions remain open for future investigation. First, the *mechanisms driving expert specialisation* deserve deeper study: why do experts converge to their particular feature combinations rather than others, and do load-balancing losses or model capacity constraints force redundancy across experts? Understanding whether experts cluster based on input similarity or functional similarity—that is, whether co-routed tokens are co-routed because they share semantic content or because they require similar downstream transformations— could reveal fundamental principles of how modular architectures self-organise.

Second, tracking how routing evolves across *training checkpoints* could reveal when and how expert specialisation emerges and how early training shapes routing. This problem could be attacked similarly to the Developmental Interpretability literature (Wang et al., 2024; Kangaslahti et al., 2025); for example, Wang et al. (2025) study the specialisation of attention heads over training.

RouterInterp takes a step toward making expert routing a transparent and controllable component of foundation models. As MoE architectures continue to power frontier systems, tools that reveal the logic behind routing decisions will be essential—not only for scientific understanding, but for ensuring these systems remain aligned with human intent.

# Acknowledgments

Thanks to Ivaylo Dimitrov for contributions to an earlier version of this project. Thanks to Cameron Holmes, Kamal Maher, Simon Schrader, and Aran Arslan for helpful comments on earlier drafts of this work. Thanks to David Africa, Arathi Mani, Edmund Lau, Sid Black, Nora Belrose, Goncalo Paulo, Lucia Quirke, Herbie Bradley, Andrew Gritsevskiy, Derik Kaufmann, Lucius Bushnaq, Daniel Filan, Hans Gundlach, Clément Dumas, and Peter Knees for helpful conversations. We are grateful to the SPAR Programme for additional facilitation of this project. Thanks to Cameron Holmes and Benjamin Hilton for additional support.

The authors acknowledge the use of resources provided by the Isambard-AI National AI Research Resource (AIRR). Isambard-AI is operated by the University of Bristol and is funded by the UK Government's Department for Science, Innovation and Technology (DSIT) via UK Research and Innovation; and the Science and Technology Facilities Council [ST/AIRR/I-A-I/1023]. IL was funded in part by the Austrian Science Fund (FWF) 10.55776/COE12.

# Impact Statement

We present a new method for interpreting expert routing in Sparse Mixture-of-Experts (MoE) models. When combined with existing techniques for interpreting attention, MLPs, and intermediate representations, this work can help researchers understand the inner workings of neural networks.

We anticipate that the overall effect of this work will be to accelerate progress in mechanistic interpretability and consequently improve our ability to explain and steer model behavior. Though we acknowledge the potential dual use nature of interpretability research (as with all Machine Learning research), we expect that the main applications of this work will be in debugging neural networks, improving the trustworthiness of AI systems, enabling the evaluation of fairness and bias in model decision-making, and understanding and mitigating potential risks from AI systems.

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

# A. Related Work

## A.1. Sparse Feature Decompositions for Model Interpretability

SAEs rely on the Linear Representation Hypothesis (Mikolov et al., 2013; Bricken et al., 2023; Olah et al., 2020; Nanda et al., 2023; Park et al., 2024): the hypothesis that features in LMs are represented linearly (Bereska & Gavves, 2024). The LRH is often a controversial assumption within interpretability (Smith, 2024; Csordás et al., 2024). However, in our case of interpreting expert routing, routing is usually performed by a linear map and so only linearly accessible information can be important for routing. This makes SAEs uniquely well suited to our problem[18].

SAEs have drawbacks, however. SAEs can struggle to capture logical hierarchical structure (Costa et al., 2026); they do not efficiently capture structures of multi-dimensional features (Engels et al., 2025); they do not generally capture relations across multiple tokens (Lubana et al., 2026); and the structure of Relational Composition (Wattenberg & Viégas, 2024) remains difficult for SAEs.

Most work interpreting representations to date has focused on the impact of intermediate representations on the output logits of a forward pass (Bricken et al., 2023) or on representations in a subsequent layer (Marks et al., 2025). We instead focus on how representations impact the behaviour of a subsequent routing layer.

Inspired by the apparent success of SAE-based explanations, Sharkey (2024) describes a framework for the continual improvement of MI explanations. The framework's three stages are: (1) *mathematical description* (breaking down the neural network into functional parts [19]), (2) *semantic description* (labelling each functional part in a way that is understandable to humans [20]) and (3) *validation* (using the semantic description to make predictions about model behaviour and evaluating these predictions). Here our combination of the trained SAEs and our Ansatz (conjecture) that router weights are linear combinations of weights provide our mathematical description of routing. We leverage AutoInterp (Paulo et al., 2025) to aid with semantic description and evaluation of the functional parts and the empirical success of this approach provides us with evidence for the Ansatz.

---

[18]Also note that the Tuned Lens (Belrose et al., 2023) also validates that linear probes on intermediate residual stream states capture meaningful structure.

[19]where here the parts could be in terms of representations (Bricken et al., 2023) or computations (Braun et al., 2025)

[20]note that implicitly this formulation requires the Independent Additivity Principle (Ayonrinde et al., 2024) - if multiple parts are relevant at one time, then it must be the case that we can understand the whole simply from understanding the constituent parts

## A.2. Mixture of Experts Interpretability

Shazeer et al. (2017) designed the modern Sparse MoE layer to improve the efficiency and scalability of very large neural networks. Other researchers, however, hoped that the specialisation routing provided might also extend some interpretability benefits (Jacobs et al., 1991; Fedus et al., 2022a). However, Jiang et al. (2024) find it difficult to obtain clean interpretability results with their MoE model Mixtral, remarking "Surprisingly, we do not observe obvious patterns in the assignment of experts based on the topic [of the input text]." Lewis et al. (2021) and Zoph et al. (2022) analyse the unigram patterns of routing and find that the observed level of specialisation varies dramatically making interpretability difficult. Tigges (2025) report some success with unigram analysis of a few experts such as a 'business' expert. We show, however, that by using SAE features as a basis for explanation rather than tokens, we are able to achieve more accurate and concise explanations.

Yang et al. (2025b) suggest that with very large numbers of experts, routing can be made somewhat interpretable. However, their setting requires more experts than is compute optimal given scaling laws for sparse models (Abnar et al., 2025; Ludziejewski et al., 2024) or more experts than is typical in high performing open source MoE models (Liu et al., 2024; Yang et al., 2025a; Agarwal et al., 2025; Muennighoff et al., 2025).

Chaudhari et al. (2025) provide an alternative model for understanding the phenomena of superposition (Elhage et al., 2022) in MoE models. Chaudhari et al. (2025) find that individual experts exhibit greater monosemanticity than equivalent dense models (in a toy setting). Crucially, their analysis measures monosemanticity at the level of *expert weight matrices*, not at the level of *routing decisions*, which we focus on. We argue that even if individual experts represent their assigned features monosemantically, the routing decision itself may be polysemantic. The router partitions the input space such that features assigned to the same expert only compete with each other for representational capacity (Chaudhari et al., 2025). This creates an incentive to route *dissimilar* domains to the same expert: micro-domains that rarely co-occur can share an expert with little interference, enabling the expert to effectively perform Computation in Superposition (Hänni et al., 2024).

## A.3. Adaptive Computation

Sparse MoE models are a type of Adaptive Computation model (Graves, 2016; Ayonrinde, 2023b)[21]. Adaptive Computation models are typically either more FLOP or parameter efficient than dense models as they either use a subset

---

[21]Also known as Conditional or Dynamic Computation models (Han et al., 2021)

of their parameters (for example sparse MoE models or Early Exit models (Banino et al., 2021)) or reuse parameters (for example looping models (Dehghani et al., 2019; Yang et al., 2024)) respectively. There has been little work on the interpretability of Adaptive Computation models and while our work focuses on interpreting routing in the sparse MoE layer, we would be excited about work generalising this approach to routing in early-exit models like Mixture of Depths (Raposo et al., 2024).

## A.4. Modularity in Neural Networks

Many brain-inspired neural network architectures employ modularity (the organisation of a system into functional, sparsely connected subunits) as a core component (Pfeiffer et al., 2023), as modularity is believed to be one of the key properties that makes brains efficient and effective (Clune et al., 2013).

We might hypothesise that because neural networks generalise so well and have much lower effective dimensionality than the model dimension would suggest (Lau et al., 2025), we should expect the models to contain intrinsic modularity. That is to say that even though NNs look fully connected and highly entangled, perhaps there is some way of viewing computation such that the computation is in fact highly modularised (Bushnaq et al., 2024). However, efforts to find such modularity have proven difficult (Filan et al., 2021), partially because it is not clear in what form we should expect such modularity to appear.

One way to understand the modularity that neural networks naturally and implicitly form, is to enforce some modular computation and see how the neural networks react. Sparse MoE models have this kind of enforced modularity and we hope that in studying these models we can develop better tools for uncovering possible latent modularity in dense neural networks. In particular, one promising path for understanding modularity may be through the optimisation pressures that resulted in such modularity. For example specialisation and reducing connection costs are two evolutionary pressures for modularity to develop in biological models (Clune et al., 2013; McDougall et al., 2022). Analogously, we might expect that specialisation pressures in the training processes for AI systems also results in the development of modularity as in Wang et al. (2025).

## B. Routing Depends on Multiple Features

In Section 3, we reported routing prediction results using $m = 32$ active features. Here we ablate the effect of the number of active features $m$ on routing prediction macro-F1. Table 3 shows macro-F1 for varying $m \in \{1, 8, 16, 32, 64, 128\}$ on gpt-oss-20b with SAE sparsity $s = 128$.

|  | **Layer 4** | **Layer 8** | **Layer 12** | **Layer 16** | **Layer 20** |
|---|---|---|---|---|---|
| Unigram Baseline | 0.472 | 0.442 | 0.296 | 0.343 | 0.369 |
| Bigram Baseline | 0.535 | 0.511 | 0.356 | 0.406 | 0.411 |
| SAE Predictor ($m = 1$) | 0.183 | 0.260 | 0.185 | 0.089 | 0.280 |
| SAE Predictor ($m = 8$) | 0.235 | 0.380 | 0.267 | 0.249 | 0.617 |
| SAE Predictor ($m = 16$) | 0.349 | 0.408 | 0.434 | 0.387 | 0.694 |
| SAE Predictor ($m = 32$) | 0.490 | 0.640 | 0.632 | 0.414 | 0.749 |
| SAE Predictor ($m = 64$) | **0.576** | 0.699 | 0.694 | 0.400 | 0.777 |
| SAE Predictor ($m = 128$) | 0.569 | **0.702** | **0.730** | **0.439** | **0.789** |

*Table 3.* Routing prediction improves significantly as more SAE features are used, consistent with polysemantic expert behaviour predicted by the SSH. Results are shown for gpt-oss-20b with SAE sparsity $s = 128$. Performance continues improving up to $m = 128$ in the reported layers, indicating that routing depends on multiple distinct concepts rather than a single domain.

The results reveal a clear pattern: macro-F1 generally improves as $m$ increases in both layers. Layer 16 rises from 0.089 ($m = 1$) to 0.439 ($m = 128$), and Layer 20 rises from 0.280 ($m = 1$) to 0.789 ($m = 128$). This trend indicates that routing decisions depend on multiple distinct concepts, consistent with polysemantic expert behaviour.

## C. Feature Selection Method and Set Size

In Section 3, we describe selecting the top-$n$ features by $\rho$-usefulness for each expert. Here we justify this choice by evaluating (1) how $\rho$-usefulness compares to an alternative selection method based on cosine similarity, and (2) how the feature set size $n$ affects routing prediction performance.

As an alternative to $\rho$-usefulness, we consider selecting features based on the geometric alignment between SAE decoder directions $\boldsymbol{d}_f \in W_{dec}$ and router weight vectors $\boldsymbol{w}_k \in \boldsymbol{W}_r$. For each expert $E_k$, we rank features $f$ by $\cos(\boldsymbol{d}_f, \boldsymbol{w}_k)$ and select the top-$n$.

To evaluate which feature selection method is better, we ask: which features selected by each method are most predictive of expert routing? We use the same probe setup as in Section 3, except we restrict each probe to the top-$n$ features chosen by the selection method: we train only on tokens where at least one of those $n$ features is active, setting all other non-selected features to zero. This focuses training on the most informative examples for each expert and provides a more controlled comparison of the two selection criteria.

Figure 4 shows results across five layers of gpt-oss-20b. Individual $\rho$-useful features carry stronger predictive signal than cosine-similarity features at every layer, particularly at small $n$. As the feature set grows, both methods converge to similar performance, and the mean across layers shows a plateau beginning around $n \approx 30$ features. This suggests that routing-relevant information is redundantly encoded across many SAE features, and with enough features both criteria eventually capture sufficient predictive signal.

Despite this plateau, we select $n = 10$ features per expert throughout our RouterInterp experiments. This is a deliberate tradeoff between predictive coverage and explainability: including substantially more features would produce very long natural-language expert descriptions, reducing interpretability for human readers without a commensurate gain in predictive performance (Sharkey, 2024; Ayonrinde et al., 2024; Ayonrinde & Jaburi, 2025).

## D. Causal Effect of $\rho$-useful Features

The $\rho$-usefulness score measures how discriminative a feature is for expert routing, but high correlation with routing does not by itself imply that the feature *causally* drives it. To test whether $\rho$-useful features can directly influence routing decisions, we performed a causal intervention experiment following. For each of the 64 experts in OLMoE (layer 10), we selected the 10 top $\rho$-useful features and 10 random SAE features as controls. Following Templeton et al. (2024); Arad et al. (2025), we *clamped* the feature's SAE activation to $\alpha$ times its maximum activation across the dataset at the SAE hookpoint before re-running the forward pass. We ran two complementary interventions:

- **Positive steering** ($\alpha = 1$): On tokens where the target expert was *not* in the router top-$k$ during the clean forward pass, we measured whether clamping the feature to its max activation pushed the expert into the top-$k$.

- **Negative steering** ($\alpha = -1$): On tokens where the target expert *was* in the router top-$k$ during the clean pass, we measured whether clamping the feature to the negative of its max activation knocked the expert out of the top-$k$.

Our primary metric is the change in the fraction of intervened tokens for which the expert appears in the router top-$k$, reported in percentage points (pp). Averaged across all 64 experts in the layer, $\rho$-useful features increase expert

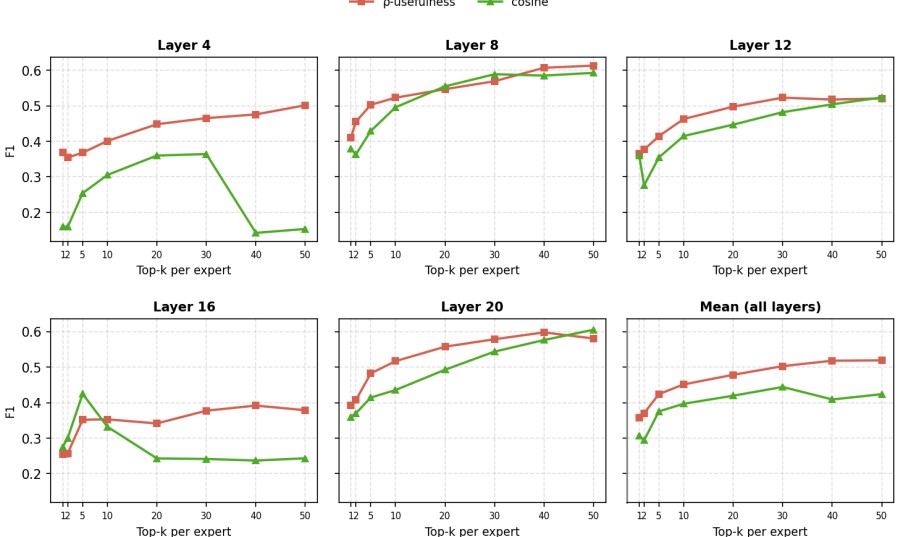

*Figure 4.* Features ranked by $\rho$-usefulness are individually more predictive of expert routing than features selected by cosine similarity with router weights, consistently across layers of gpt-oss-20b. Performance plateaus at around $n \approx 30$ features for both criteria (bottom-right panel), indicating that routing-relevant information is redundantly encoded across many SAE features.

activation by $+13.5$ pp at $\alpha = 1$ and decrease it by $-40.9$ pp at $\alpha = -1$. The corresponding gaps over the random-feature controls are $+12.3$ pp at $\alpha = 1$ and $-29.4$ pp at $\alpha = -1$, and the per-expert mean $\rho$-useful effect exceeds the per-expert mean random-feature effect (in the expected direction) for $59/64$ experts at $\alpha = 1$ and $61/64$ experts at $\alpha = -1$ ($\sim$2M tokens). This provides causal evidence that $\rho$-useful SAE features are not merely correlated with routing decisions but can directly influence them.

## E. Routing Prediction Across Layers

We extend the matched-sparsity probe comparison of Section 3 to five layers of gpt-oss-20b at two sparsity levels, $s \in \{64, 128\}$. Table 4 reports macro-F1 for SAE, neuron, and PCA sparse linear probes. At $s{=}128$, the SAE predictor outperforms the best neuron and PCA probe at four of five layers, showing that the SAE-basis advantage holds across both depth and sparsity. We compare two SAE variants ($s{=}64$ and $s{=}128$); the effect of the underlying SAE is discussed in Section G.

## F. Evidence for Superposed Specialisation

This appendix provides complementary tests of the *Superposed Specialisation Hypothesis* (SSH) that supplement the analysis of Section 4.3. Each test takes a different angle on the same question: does each expert's routing correspond to a single coherent macro-domain (DSH), or to a disjoint union of micro-domains (SSH)?

### F.1. Feature Co-activation Analysis

The instantiation of sim in Section 4.3 measures *semantic* relatedness between latents via their natural-language explanations. Here we re-run the same clustering test under a *contextual* similarity: how often two latents fire on overlapping token contexts. The two notions are independent axes of relatedness, and the SSH predicts dissimilarity on both.

**NPMI as co-firing similarity.** Following Jiang et al. (2025), who use normalised pointwise mutual information (NPMI) between SAE latents for SAE-based data analysis, we measure co-firing as

$$\text{NPMI}(f_j, f_k) = \frac{\log P(f_j, f_k) - \log P(f_j)P(f_k)}{-\log P(f_j, f_k)} \in [-1, 1],$$
(6)

where $P(f_j)$ is the fraction of tokens at which latent $f_j$ fires, estimated over $\sim 10^7$ tokens. $\text{NPMI} = 0$ means statistical independence (firing rates uncorrelated), positive values indicate excess co-firing, and negative values indicate co-firing below the chance rate. We define the co-firing similarity

$$\text{sim}_{\text{NPMI}}(f_j, f_k) = \max\big(0, \text{NPMI}(f_j, f_k)\big),$$
(7)

clamping the negative side to 0 (latents that fire below chance together are no more "similar" than independent latents for the purpose of grouping micro-domains).

**Clustering.** We then mirror the pipeline of Section 4.3 but replace sentence-embedding $k$-means with *hierarchical* clustering on the distance $1 - \text{sim}_{\text{NPMI}}$. As in Section 4.3, we

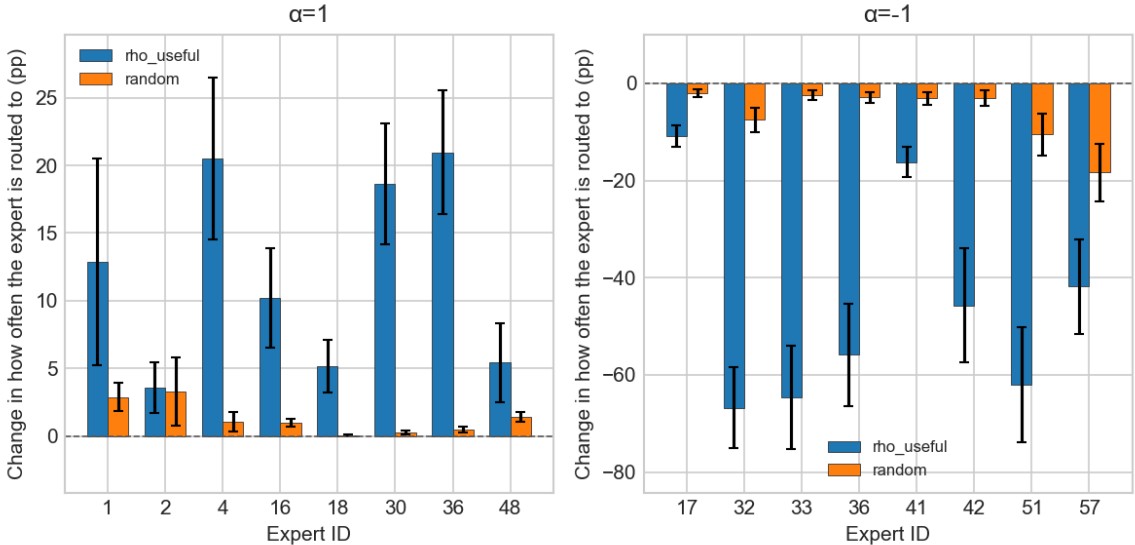

*Figure 5.* Interventions along directions of $\rho$-useful features change how often the target expert is routed to more strongly than interventions along random SAE features. Each bar is the percentage-point change in the fraction of intervened tokens for which the expert is in the router top-$k$, between the clean and intervened forward passes (mean ± SEM over 10 features per expert). For positive steering ($\alpha = 1$), we intervene on tokens where the expert was *not* in the router top-$k$ on the clean pass; for negative steering ($\alpha = -1$), we intervene on tokens where the expert *was* in the router top-$k$ on the clean pass. Results are for OLMoE, layer 10, evaluated on 2M tokens.

|  | **Layer 4** | **Layer 8** | **Layer 12** | **Layer 16** | **Layer 20** |
|---|---|---|---|---|---|
| SAE Predictor ($s=128$) | **0.569** | 0.702 | **0.730** | 0.439 | **0.789** |
| SAE Predictor ($s=64$) | 0.552 | **0.725** | 0.685 | 0.297 | 0.773 |
| Neuron Probe ($s=128$) | 0.519 | 0.638 | 0.586 | **0.559** | 0.630 |
| Neuron Probe ($s=64$) | 0.476 | 0.603 | 0.509 | 0.524 | 0.570 |
| PCA Probe ($s=128$) | 0.544 | 0.623 | 0.536 | 0.494 | 0.366 |
| PCA Probe ($s=64$) | 0.517 | 0.603 | 0.523 | 0.482 | 0.384 |

*Table 4.* The learned SAE basis captures more routing-relevant structure than standard bases at matched sparsity across most layers of gpt-oss-20b. All methods use sparse linear probes with matched sparsity ($s$ active dimensions). At $s=128$, the SAE predictor achieves strictly higher macro-F1 than the best neuron or PCA probe at four of five layers. All values reported as macro-F1.

cluster all $E \times n$ layer-wide latents globally and count for each expert how many of these clusters its $n$ latents occupy ($G^{\mathrm{NPMI}}(E_i)$).

**Effect of $n$.** We additionally raise the per-expert feature set size from $n=10$ to $n=20$ to check that the conclusion is not an artefact of an aggressive top-$n$ cut.

**Result.** Figure 6 shows that $G^{\mathrm{NPMI}}$ remains close to the SSH upper bound under the contextual similarity: $17.2/20$ for gpt-oss-20b and $12.9/20$ for OLMoE-1B-7B. The conclusion is therefore robust to (i) the choice of similarity, and (ii) the per-expert feature set size, with the NPMI variant if anything closer to the SSH prediction of $G=n$ on gpt-oss-20b (86% of $n$) than the semantic variant (78% of $n$).

We also report the underlying NPMI values directly. The

mean within-expert pairwise NPMI is $0.052$ on gpt-oss-20b (layer 19) and $-0.0004$ on OLMoE-1B-7B (layer 14)—both near 0 (statistical independence) and far from the DSH expectation of NPMI close to 1 for latents describing a single coherent macro-domain. Within-expert latents therefore fire in essentially independent contexts.

### F.2. The Pile Case Study

If the DSH holds, experts should preferentially route tokens from particular corpus domains; if the SSH holds, each expert should draw tokens from a broad mix of domains. (Jiang et al., 2024) performed a similar analysis for Mixtral, plotting the proportion of tokens from different domains assigned to each expert, but found no obvious topic-based patterns. We follow their approach using the Pile dataset (Gao et al., 2020), which comprises di-

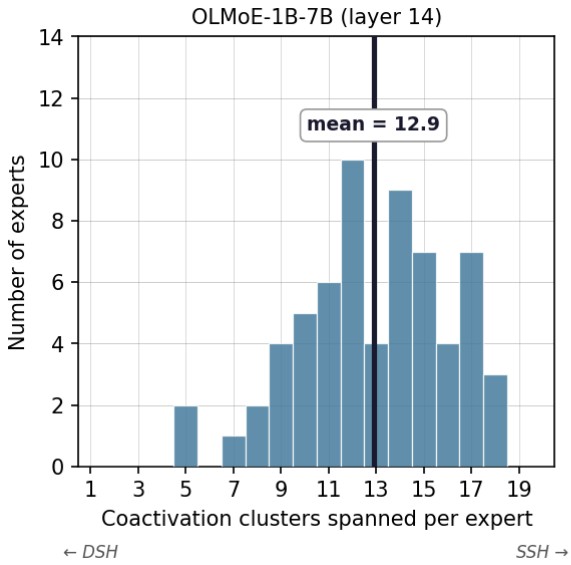
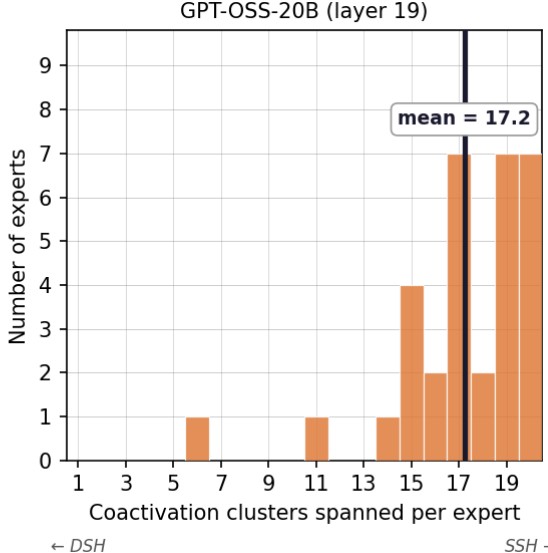

*Figure 6.* **Routing-predictive latents are contextually disjoint: nearly every latent that best predicts an expert's activation fires in its own context cluster.** For each expert we take its top $n{=}20$ $\rho$-useful SAE latents $\mathbb{F}_i$, and cluster all $E \times n$ layer-wide latents by their co-firing similarity $\text{sim}_{\text{NPMI}}$. We then count how many of these clusters, $G^{\text{NPMI}}(E_i)$, each expert's $n$ latents occupy. The DSH predicts $G^{\text{NPMI}}(E_i){=}1$ (all of an expert's latents fall in a single cluster), whereas the SSH predicts $G^{\text{NPMI}}(E_i){=}n$ (each latent occupies its own cluster). The empirical values are consistent with the SSH prediction in both models: mean $G^{\text{NPMI}}$ is 17.2 on gpt-oss-20b (layer 19) and 12.9 on OLMoE-1B-7B (layer 14).

verse subsets—including ArXiv, PubMed, GitHub, StackExchange, Wikipedia, and FreeLaw, among others—spanning scientific, legal, code, and web domains. We process ~10M tokens over $N_{\text{subsets}} = 16$ Pile subsets. For each expert $E_i$, we compute the proportion of tokens from each subset that are routed to $E_i$, yielding a per-expert distribution over subsets.

We quantify domain specialisation via **normalised entropy**: the Shannon entropy (Shannon, 1948) of each expert's subset distribution divided by $\log N_{\text{subsets}}$, so that 1.0 corresponds to a perfectly uniform distribution over all subsets (no specialisation) and 0.0 corresponds to routing from a single subset only (maximum specialisation).

**Corpus-proportional baseline.** The Pile subsets are not equally sized, so even an expert with no domain preference will not have an entropy of exactly 1.0. The appropriate reference baseline is the *corpus-proportional* distribution: the entropy an expert would exhibit if it mirrored the global mix of Pile subsets exactly, i.e. if its per-subset routing shares were proportional to the corpus-wide token counts. This baseline evaluates to 0.8 (normalised); any expert whose entropy matches this value is simply reflecting the composition of the corpus rather than specialising.

As shown in Figure 7, expert entropies cluster tightly around the corpus-proportional baseline across all layers, with very few experts deviating substantially. This confirms that ex-

perts route tokens from all Pile subsets in approximately the same proportions as the corpus overall.

## G. SAE Training Details

Following Section 3, we report SAE training details and reconstruction quality. SAEs are trained on residual-stream activations before each layer's attention block.

**OLMoE-1B-7B.** We train Top-K SAEs using the Sparsify library on activations from layers 3, 7, 11, and 15 of OLMoE-1B-7B, sampled from 100M tokens of OLMoE-mix-0924 (Muennighoff et al., 2025). Each SAE has 32,768 features (16× expansion) with per-token latent sparsity $s{=}32$. Training all four SAEs took approximately 2 hours in total on a single A100 GPU. Table 5 reports the fraction of variance unexplained (FVU) and the percentage of dead features (features that never activate on the evaluation set).

| Layer | FVU | Dead features (%) |
|-------|-------|-------------------|
| 3 | 0.007 | 4.11 |
| 7 | 0.024 | 0.51 |
| 11 | 0.064 | 0.59 |
| 15 | 0.218 | 0.51 |

*Table 5.* SAE reconstruction quality for OLMoE-1B-7B (Top-K; 32,768 features, 16× expansion, $s{=}32$). FVU increases with depth while dead feature rates remain low.

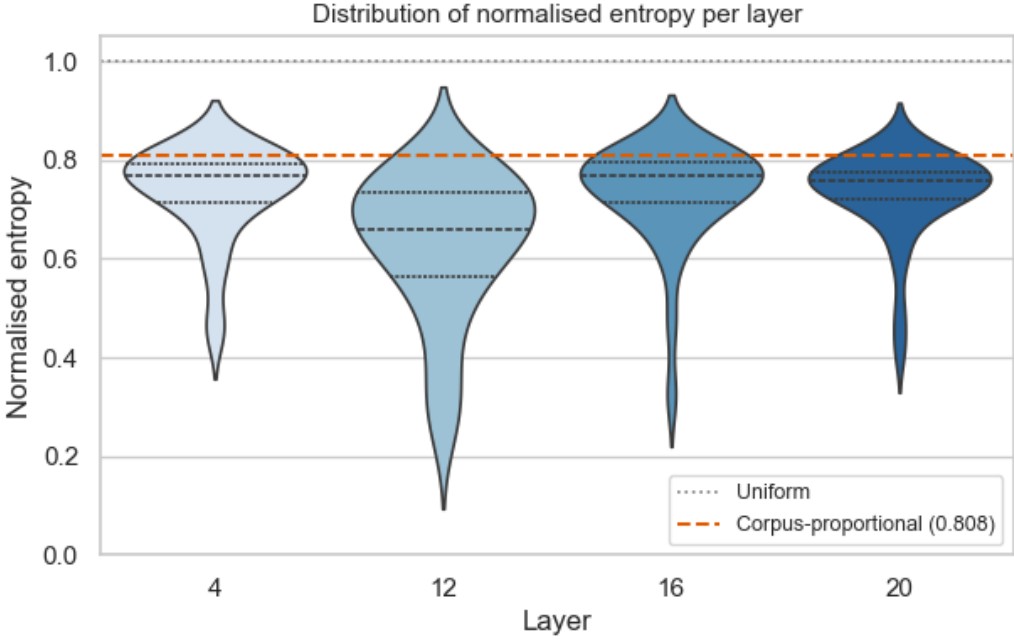

*Figure 7.* Experts show little macro-domain specialisation across 16 Pile subsets. Each violin shows the distribution of normalised entropy of each expert's Pile subset routing distribution across 32 experts at a given layer of gpt-oss-20b. The dashed line marks the corpus-proportional baseline (0.8): the entropy an expert would have if it simply mirrored the unequal sizes of the 16 Pile subsets. Expert entropies cluster near this baseline at every layer, indicating that routing reflects corpus composition rather than domain preference.

**gpt-oss-20b.** For gpt-oss-20b, we use publicly available BatchTopK SAEs from Lin (2025),[22] trained on layers 4, 8, 12, 16, and 20. Each SAE has 131,072 features ($\sim$45× expansion) with two per-token latent sparsity variants: $s$=64 and $s$=128. Table 6 reports FVU and dead feature rates for both variants.

| | FVU | | Dead features (%) | |
|---|---|---|---|---|
| **Layer** | $s$=64 | $s$=128 | $s$=64 | $s$=128 |
| 4 | 0.050 | 0.037 | 0.04 | 0.90 |
| 8 | 0.101 | 0.077 | 0.02 | 0.29 |
| 12 | 0.172 | 0.137 | 0.00 | 0.01 |
| 16 | 0.232 | 0.191 | 0.00 | 0.00 |
| 20 | 0.146 | 0.117 | 0.02 | 0.00 |

*Table 6.* SAE reconstruction quality for gpt-oss-20b (BatchTopK; 131,072 features, $\sim$45× expansion). FVU peaks at layer 16 for both sparsity levels; $s$=128 consistently achieves lower FVU than $s$=64.

**RouterInterp quality is downstream of SAE quality.** Across tasks and layers, SAE reconstruction quality correlates with RouterInterp's downstream performance. This is most visible at layer 16 of gpt-oss-20b, where FVU is

[22]https://huggingface.co/andyrdt/saes-gpt-oss-20b

highest (0.232 for $s$=64, 0.191 for $s$=128) and both routing prediction macro-F1 (0.297 for $s$=64; 0.421 for $s$=128) and explanation scores (0.53 for $s$=64; 0.56 for $s$=128) are weakest. Improving SAE quality at this layer—FVU drops from 0.232 to 0.191 when moving from $s$=64 to $s$=128—yields the largest gain in routing prediction (0.297 $\rightarrow$ 0.421). The consistent co-occurrence of high FVU and weak downstream scores confirms that poor reconstruction is a bottleneck: RouterInterp's performance is bounded by SAE quality, and better SAEs would improve RouterInterp's performance.

## H. Human Evaluation

In Section 5.1, we introduced an LLM-based scorer to evaluate RouterInterp explanation quality as a proxy for human judgment. To validate whether this LLM scorer faithfully reflects human judgment, we conducted a blind human evaluation in which human raters performed the same binary labeling task as the LLM scorer, allowing us to measure how consistently human raters agree with each other (inter-annotator agreement) and whether the LLM scorer matches human judgments. We collected annotations on 150 text/context pairs (6 examples per expert across 25 experts) from layer 11 of gpt-oss-20b. Each example was labeled by two independent human evaluators and by the LLM scorer. We report pooled agreement and Cohen's $\kappa$ with uncertainty

estimated over repeated resampling.

| Metric | Human–Human | Human–LLM |
|---|---|---|
| Pooled agreement | $0.807 \pm 0.070$ | $0.857 \pm 0.050$ |
| Cohen's $\kappa$ | $0.487 \pm 0.165$ | $0.602 \pm 0.150$ |

*Table 7.* The LLM scorer is at least as consistent a judge as a human rater. Pooled agreement is the fraction of examples where both raters gave the same label; Cohen's $\kappa$ is a stricter metric that accounts for the fact that two raters could coincidentally agree just by both labelling most examples the same way. Values are mean $\pm$ std over repeated resampling; measured on 150 examples across 25 experts from layer 11 of gpt-oss-20b.

Human–LLM agreement is comparable to Human–Human agreement on both metrics, with no statistically significant difference. Because only two human annotators were compared per example, some disagreement is expected from subjective thresholds on edge cases. Despite this limited study scale, the results support using our LLM scorer as a practical and reliable proxy for human interpretability judgments.

Our findings are also consistent with prior evidence that automated interpretation and LLM-as-a-judge methods correlate with human raters and can scale beyond manual-only pipelines (Paulo et al., 2025; Zheng et al., 2023; Anthis et al., 2025).

## I. Feature Sharing

In RouterInterp we explain each expert through its top-10 $\rho$-useful SAE features (Section 3). Here we ask how much these top features overlap across experts in gpt-oss-20b, and what that overlap implies for DSH.

**Most top-$\rho$ features are shared.** On average only 18.4% of each expert's top-10 features are unique to that expert; the other 81.6% also appear in the top-10 of at least one other expert (Figure 8a). The pool of routing-relevant features is small: across all $32 \times 10$ slots, only 62–138 distinct SAE features fill them, depending on layer.

**Implication for expert specialisation.** DSH posits that each expert specialises in a single semantically coherent domain. The only way DSH can produce 81.6% feature sharing (Figure 8a) is if many experts cover the *same* coherent domain. In that case, the features they share would describe that common domain and should themselves form a coherent set. But Section 4.3 shows that each expert's top features already span many disjoint semantic clusters ($G(E_i) \approx$ 7–8 out of 10). The shared features are therefore not coherent within a single expert, and there is no reason to expect coherence across experts either.

The rank profile (Figure 8b) reinforces this conclusion. Un-

| Layer | Macro-F1 | Spearman $\rho$ | MSE |
|---|---|---|---|
| 4 | 0.575 | 0.258 | 0.0278 |
| 8 | 0.708 | 0.809 | 0.0294 |
| 12 | 0.729 | 0.741 | 0.0295 |
| 16 | 0.421 | 0.413 | 0.0285 |
| 20 | 0.796 | 0.817 | 0.0296 |

*Table 8.* The predicted probabilities from SAE-based probes align well with the actual MoE router probabilities, even though the probes are trained only with binary cross-entropy on top-$k$ membership. Spearman $\rho$ is the rank correlation between the predicted and true probability assigned to each expert, computed per token and averaged; high values mean the probe recovers which experts the router assigns more probability to. MSE is the mean squared error between predicted and true per-expert probabilities ($\approx 0.03$; RMSE $\approx 0.17$). Spearman correlates with macro-F1 across layers (both strong at 8, 12, 20; weaker at 4 and 16). gpt-oss-20b; SAE predictor with $s=128$.

der DSH, the rank-1 $\rho$-useful feature should be the strongest signal for an expert's coherent domain, and therefore the most expert-distinguishing. Uniqueness should be highest at the top ranks and fade at peripheral ranks. We observe the opposite. Uniqueness is lowest at rank 1 and rises monotonically to 40–60% by rank 10. The features that most strongly predict routing are precisely the most widely shared, while expert-specific features are confined to ranks where they barely influence the router. Together these observations are hard to reconcile with any reading of DSH.

**Implication for RouterInterp.** The fact that many experts share top-$\rho$ features also motivates the hard-negative construction in RouterInterp's scoring setup (Section 5.1): for each context routed to $E_i$, we draw hard negatives from contexts routed to some $E_j$ with $\mathbb{F}_i \cap \mathbb{F}_j \neq \varnothing$ — contexts that look similar in feature space but the router still routes elsewhere.

## J. Routing Prediction Correlation

In Section 3, we evaluated SAE-based routing prediction using Macro-F1. As an additional validation of the SAE predictor's capabilities, we check whether the predicted probabilities also align with the router's continuous output. The classifiers are trained only with binary cross-entropy on top-$k$ membership and are not directly optimised to recover the full probability distribution. For each token, we take the vector of predicted probabilities and the vector of true router probabilities (one entry per expert) and compute their Spearman rank correlation and MSE.

Table 8 shows that at layers where classification is strong (8, 12, 20), Spearman $\rho$ exceeds 0.7, indicating that the probe recovers not just which experts are selected but their relative ranking. At layers 4 and 16, where macro-F1 is weaker, calibration drops accordingly. MSE is uniformly low ($\approx 0.03$;

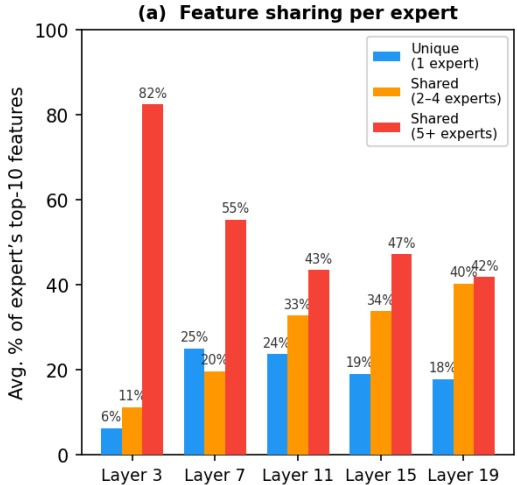

*Figure 8.* Top-$\rho$ features are heavily shared across experts in gpt-oss-20b. (a) Each expert's top-$\rho$ features are dominated by features shared across many other experts rather than by expert-specific ones: only 18.4% are unique to an expert, while 81.6% appear in the top-10 of at least one other expert. DSH would only account for this if many experts covered the same coherent domain, but Section 4.3 shows that within a single expert the top features already span $\sim$7–8 disjoint clusters out of 10. (b) The features that most strongly predict routing are the most widely shared rather than expert-specific, the opposite of what DSH predicts: uniqueness is lowest at rank 1 and rises monotonically with rank (40–60% by rank 10).

RMSE $\approx 0.17$), showing that predicted probabilities stay close to the true router values even at weaker layers.

## K. Example Expert Explanations

Table 9 shows different levels of RouterInterp explanations for a randomly selected expert (Expert 2) from layer 11 of gpt-oss-20b. It illustrates how expert-level explanations aggregate multiple feature-level explanations, which are in turn acquired by explaining examples where the feature activates.

## L. Prompts

We provide the prompts used for expert-level explanations, the n-gram explanation baseline, and automatic explanation scoring (Section 5.1).

**RouterInterp.** To synthesise an expert-level description from numbered per-feature explanations, the model receives the following instruction, followed by the list of feature explanations and the requested output format.

```
    You summarise neural network feature
explanations into a description for an
MoE expert.

    These features are connected via OR
logic — the expert activates when ANY of
these conditions is met. Experts are
polysemantic and fire in multiple
```

```
distinct, unrelated contexts.

    Your summary will be used by a
classifier to decide whether text
examples match the expert's activation
pattern. Both coverage and precision
matter: missing a condition means missed
activations, but unqualified broad
conditions cause false matches.

    KEY PRINCIPLE — QUALIFY EVERY
CONDITION:
    Each condition should name a pattern
AND give concrete examples that help a
classifier decide yes/no. Never use a
broad category without qualifying
examples, and never list only isolated
tokens without the broader pattern they
belong to.

    Some feature explanations will be
broad because features are sampled across
 activation quantiles. These carry real
domain signal --- use them to provide
context for specific features, combining
both into well-qualified conditions.

    Your summary must:
    - Preserve ALL distinct activation
conditions (the expert fires for ANY of
them)
    - Qualify broad conditions with
specific examples, tokens, or sub-
patterns
```

```
     - Group related conditions together,
but do NOT merge distinct ones
     - Be comprehensive - missing a
condition means missing when the expert
fires
     - Be dense - describe conditions
concisely without losing details or
padding
     - Do not reference feature numbers or
 indices
     - Write in flowing prose paragraphs,
NOT numbered or bulleted lists

     {n_features} feature explanations for
 one expert (connected by OR - fires if
ANY match):

     {explanations}

     Write a concise prose summary of ALL
activation conditions. For each condition
, include the broad pattern AND specific
qualifying examples. Group related
conditions into paragraphs. Do NOT use
numbered lists or bullet points. Output
ONLY the summary.
```

**Expert activations AutoInterp.**  The baseline that explains routing from expert-activating text without SAE decomposition follows the standard AutoInterp explainer pattern: a system instruction, fixed few-shot user and assistant turns that establish the span-marking convention (omitted here), and a final user message built from the target expert's activating training passages.

```
You are a meticulous AI researcher
conducting an important investigation
into patterns found in language. Your
task is to analyze text and provide an
explanation that thoroughly encapsulates
possible patterns found in it.
Guidelines:

You will be given a list of text examples
 on which special words are selected and
between delimiters like <<this>>. If a
sequence of consecutive tokens all are
important, the entire sequence of tokens
will be contained between delimiters <<
just like this>>. When routing weights
are shown after an example, they appear
in parentheses and reflect how strongly
this expert was assigned those tokens.
The snippets you see are only samples of
contexts where the router assigned tokens
 to a particular expert in an MoE
transformer.

- Your description may need multiple
sentences or short paragraphs when many
unrelated triggers appear.
```

- Be comprehensive and dense: no padding,
 no numbered or bulleted lists, no
mention of delimiter markers (<< >>).
- Both coverage and precision matter:
missing a condition means missed
activations; vague or over-broad wording
causes false matches.
- Each distinct activation condition
should name a pattern AND give concrete
cues (examples, token types, or sub-
patterns). Do not collapse unrelated
triggers into one vague theme; preserve
them as separate conditions.
- Group related cues in the same
paragraph, but do NOT merge distinct
conditions.

End your response with a single block
introduced by [EXPLANATION]: containing
the full description (all sentences and
paragraphs of the explanation belong
after that tag).

**N-gram baseline.**  For the n-gram explanation baseline, the model is asked to compress a list of frequent co-occurring tokens (bigrams) into a single-line pattern description.

```
     Your job is to look for patterns in
text. You will be given a list of WORDS,
your task is to provide an explanation
for what pattern best describes them.
Here are some guidelines:
     - Produce a specific final
description for the latents common in the
 examples, and what patterns you found.
     - Don't focus on giving examples of
important tokens, if the examples are
uninformative, you don't need to mention
them.
     - Do not make lists of possible
explanations. Keep your explanations
short and concise.
     - The last line of your response must
 be the formatted explanation, using [
EXPLANATION]:
```

**Explanation scoring.**  For detection-style scoring, the scorer model is instructed to label each snippet against the expert explanation, after a fixed few-shot block (omitted here) that illustrates the task.

```
     You are an intelligent and meticulous
 linguistics researcher.

     You will be given a prose description
 of when an MoE expert activates ---
written so that a reader can judge new
text against it.
```

```
    Your job is to apply that description
 faithfully: decide whether each snippet
matches what the explanation claims, as
closely as the wording allows.

    You will then be given several text
snippets. For each snippet, return 1 if
it satisfies the explanation (the expert
would be described as activating there,
per the explanation), and 0 otherwise.

    Return a valid Python list only, one
entry per snippet in order. Do not return
 anything else.

    Expert activation pattern:
    {explanation}
```

| Level | Explanation |
|---|---|
| **Expert 2** | This expert activates across following distinct contexts: (1) Physical trauma and medical conditions—high activation on "bruis" and variants ("bruised", "bruises"), plus fragments like "blister", "abras", "haemat", "infarct", and "stunned". (2) Technical and structured writing—code, mathematical expressions, scientific notation, legal citations, and medical terminology from programming, mathematics, science, law, and academia, including punctuation and formatting elements. (3) Brand and term suffixes—final segments of brand names, company names, or technical terms before spaces or punctuation. (4) Electronic components—activation on "diode", "resistor", and "rectifier" in patents, diagrams, and technical specifications. |
| *Feature 41223* | *Physical trauma terms—"bruis" and variants ("bruised", "bruises"), plus fragments of injury-related terms ("blister", "abras", "haemat", "infarct", "stunned") in clinical or descriptive contexts.* **Activating Examples:** [1] suffered scr apes , bruis es , bleeding [2] . Her face was bruis ed and there |
| *Feature 71261* | *Programming and scientific notation—code snippets, structured data, and formal keywords with high activation on punctuation, symbols, and domain-specific terms.* **Activating Examples:** [1] public Flight execute () throws API [2] 77 : 58 , 78 : |
| *Feature 17613* | *Structured context punctuation—punctuation, symbols, and function words in code, citations, mathematical expressions, and legal or technical documents.* **Activating Examples:** [1] g\{} l _{ L ^ p } [2] , Y . S . P iao and |
| *Feature 1738* | *Brand and term suffixes—final segments of brand names, company names, or technical terms before spaces or punctuation.* **Activating Examples:** [1] Inverted microscope; Leica Microsystems GmbH [2] viewed using a Le itz Aristoplan microscope |
| *Feature 84536* | *Electronic component names—"diode", "resistor", "rectifier", and related abbreviations in circuits, patents, and technical specifications.* **Activating Examples:** [1] Q3, and diode D 1 . [2] stations and as rect ifiers for high-vol |

*Table 9.* RouterInterp first explains features by looking at examples where these features activate strongly, and then aggregates these explanations into an expert-level explanation. For example, Expert 2 from gpt-oss-20b layer 11 specialises in physical trauma (*Feature 41223*), programming and scientific notation (*Feature 71261*), structured punctuation (*Feature 17613*), brand-name suffixes (*Feature 1738*), and electronic components (*Feature 84536*). We show five feature-level explanations selected from the expert's top-10 $\rho$-useful features to illustrate distinct themes in the expert-level summary.

