# OpenReview forum: "RouterInterp: Understanding Superposed Specialisation in Mixture of Experts Routing"
_ICML.cc/2026/Conference — ICML 2026 regular_

### Official Review · Reviewer_L2xV · 2026-03-08

**Soundness:** 3
**Presentation:** 3
**Significance:** 3
**Originality:** 3
**Overall Recommendation:** 4
**Confidence:** 2

**Summary:**

This paper introduces RouterInterp, a method for interpreting expert routing decisions in Sparse Mixture-of-Experts (MoE) models using Sparse Autoencoders (SAEs). The authors propose two alternative hypotheses for expert specialisation: the Domain Specialisation Hypothesis (DSH), which posits that experts specialise in semantically coherent domains, and the Superposed Specialisation Hypothesis (SSH), which proposes that experts specialise in disjoint collections of unrelated micro-domains.
The authors provide theoretical arguments for the SSH based on (i) interference minimisation during in-expert computation, and (ii) load-balancing pressures during training. They develop RouterInterp, which identifies SAE features most predictive of routing via a ρ-usefulness metric that measures the difference in expected feature activation when an expert is selected versus not selected. The method then generates aggregated natural language explanations by prompting a language model with the top-n feature explanations per expert.
On gpt-oss-20b, RouterInterp achieves 81% routing prediction recall (compared to 55% for bigram baselines) and 0.62 explanation scores (compared to 0.35 for token-based explanations). The work provides both a scalable method for generating accurate explanations of MoE routing and empirical evidence supporting the Superposed Specialisation Hypothesis.

**Compliance With Llm Reviewing Policy:**

Affirmed.

**Final Justification:**

Thank you for the additional clarification and experiments. Details can be found in the last comment.

After considering both the authors’ updates and the broader reviewer discussion, I am happy to maintain my positive assessment, but I prefer to keep my score at weak accept rather than raise it further.

**Key Questions For Authors:**

1. Have you conducted any human evaluation of the generated explanations? If the explanation score improvement (0.62 vs 0.35) holds under human judgment, this would substantially strengthen the paper. How would you respond to concerns that LLM-based evaluation may not capture genuine human interpretability? [A positive response demonstrating human validation would address the primary evaluation concern and could raise my recommendation to "accept".]
2. The paper shows SAE features predict routing better than tokens, but this is correlational. Have you considered interventional experiments, e.g., ablating specific SAE features and measuring effects on routing decisions, to establish that routing operates through these features causally? [Evidence of causal relationships would significantly strengthen the support for SSH and improve the soundness rating.]
3. What are the reconstruction error and sparsity levels of the SAEs used? Given the acknowledged polysemanticity of many features, how do you distinguish "genuine superposed specialisation" from "SAE feature conflation"? Would using higher-quality SAEs (e.g., with more latents or different architectures) change the conclusions? [Clarification here would address circularity concerns and strengthen confidence in the method's validity.]
4. Several frontier MoEs use Expert Choice routing where experts select tokens rather than tokens selecting experts. Does the SSH framework and RouterInterp method apply to this setting, or would significant modifications be needed?  [This addresses generalisability concerns; a clear path to extension would increase significance.]
5. You mention that features with high $\rho$-usefulness for different experts may share features $F_i \cap F_j \neq \emptyset$. What fraction of top features are shared across experts, and what does this imply about the structure of expert specialisation? [This analysis would deepen understanding of expert relationships and could reveal additional insights about MoE organisation.]

**Limitations:**

Yes, the authors adequately discuss limitations in Section 7, including: dependency on SAE and AutoInterp quality, explanation length scaling with feature count, and evaluation limited to two models in the 1B-20B range. They appropriately call for future work on frontier architectures with shared experts and different routing methods. The Impact Statement acknowledges dual-use concerns while positioning the work's primary applications in debugging and safety.

**Strengths And Weaknesses:**

Strengths:
- Originality: The conceptual contribution distinguishing Domain Specialisation from Superposed Specialisation Hypotheses is valuable and provides a compelling explanation for why prior MoE interpretability work has struggled. The framing draws a thoughtful analogy to neural superposition (Elhage et al., 2022) and extends it to the routing level. The theoretical arguments in Section 3.2 (interference minimisation and load-balancing effects) are intuitive and well-motivated.
- Soundness: The experimental design is generally solid. The authors evaluate on two models (OLMoE-1B-7B and gpt-oss-20b) with consistent results. The use of ρ-usefulness for feature selection is principled, and the comparison against both geometric (cosine similarity) and token-based baselines strengthens the claims. The ablation in Appendix B showing performance across varying numbers of active features provides useful evidence for the polysemantic nature of routing.
- Significance: Understanding MoE routing is increasingly important as MoE architectures power frontier models. The 77% improvement in explanation accuracy over prior methods is substantial. The method is practical, it leverages existing SAE infrastructure and standard AutoInterp pipelines, and could enable better understanding of a previously opaque model component.
- Presentation: The paper is well-written with clear exposition of the hypotheses. Figure 1 provides a helpful overview, and Figure 2 effectively illustrates the distinction between DSH and SSH. The experimental sections flow logically.

Weaknesses:
- Evaluation methodology concerns: The explanation scoring (Section 4.4) relies on an LLM (Claude Haiku 4.5) as a proxy for human judgment. While this enables scalability, it introduces potential confounds. The scorer may have biases correlated with the explainer model, and the evaluation doesn't include any human validation. Even a small human study (e.g., 50-100 examples) would substantially strengthen claims about "human-understandable" explanations.
- Limited causal evidence for SSH: While the paper provides correlational evidence that SAE features predict routing better than tokens, it doesn't establish that routing operates via superposed specialisation causally. The theoretical arguments (Section 3.2) are plausible but speculative. The load-balancing argument depends on specific training dynamics (small batch sizes) that aren't verified for the evaluated models.
- SAE quality dependency: The authors acknowledge in Section 7 that many individual feature explanations are polysemantic. This raises circularity concerns: if SAE features themselves conflate multiple concepts, is RouterInterp genuinely revealing superposed specialisation, or merely reflecting SAE limitations? The paper would benefit from analysis of SAE reconstruction quality and feature monosemanticity.
- Narrow model coverage: Evaluation on only two models (1B and 20B parameters) from similar architectures limits generalisability claims. The authors note this limitation but the gap to frontier models (which often have different routing mechanisms like Expert Choice, shared experts, or very different expert counts) remains substantial.
- Incomplete baseline comparison: The n-gram baseline is a reasonable starting point, but the paper doesn't compare against other interpretability approaches for MoE models (e.g., the geometric analysis in Yang et al. 2025b, or attention-based probing). The claim of "77% higher accuracy than prior methods" conflates improvement over a simple baseline with improvement over the state of the art.

---

> ### Author Rebuttal · Authors · 2026-03-31
>
> We thank Reviewer L2xV for their thoughtful feedback. Please find our detailed responses below.
>
> **Human Evaluation of LLM-Based Explanation Scoring (Q1, W1)**
>
> We conducted a blind human evaluation with 150 text/context pairs (6 examples × 25 experts, layer 11, gpt-oss-20b), each rated by 2 annotators:
>
> |                    | Human-Human     | Human-LLM       |
> |--------------------|-----------------|-----------------|
> | Pooled Agreement   | 80.7% ± 7.0%   | 85.7% ± 5.0%   |
> | Cohen's κ          | 0.487 ± 0.165  | 0.602 ± 0.150  |
>
> Human-LLM agreement is comparable to Human-Human agreement, suggesting our LLM scorer is a valid proxy rather than exploiting model-specific confounds. Prior work supports this: Paulo et al. (2024) showed AutoInterp correlates with human raters; Anthis et al. 2025 (LLM Social Simulations Are a Promising Research Method) demonstrate LLMs serve as reliable proxies. We updated Section 4.4 and added an appendix with full details.
>
> **Causal Evidence for SSH and Load-Balancing Dynamics (Q2, W2)**
>
> We agree our evidence is correlational. We added an acknowledgment to Section 7 (Limitations) noting that interventional experiments (e.g., ablating SAE latents and measuring routing effects) would provide causal evidence and are an important future direction.
>
> Regarding batch size: OLMoE uses batch size 1024 (gpt-oss batch size unpublished). We relaxed our wording in Section 3.2 (Argument 2) and added a footnote noting this. We thank the reviewer for noting this point.
>
> **SAE Quality, Reconstruction Metrics, and Feature Conflation (Q3, W3)**
>
> We added an appendix with SAE reconstruction/sparsity details and a quality ablation. FVU and dead feature tables for both models are in the appendix (gpt-oss-20b metrics also in **Q1/Q2 of Reviewer Z96h**).
>
> We reran all gpt-oss experiments with K=128 alongside K=64. K=128 achieves uniformly lower FVU, translating into improved routing prediction (macro-F1: 0.606→0.646, **W2 of Reviewer Z96h**) and explanation scores (0.556→0.586, **Q1 of Reviewer 7XDd**). This is especially visible at layer 16, where FVU is highest (0.232) and performance weakest; K=128 (FVU 0.191) yields the largest gain (0.297→0.421). This confirms poor reconstruction is a bottleneck: better SAEs strengthen rather than undermine our conclusions.
>
> *Distinguishing genuine specialisation from SAE artifacts.* Monotonic improvement from K=64→K=128 across both tasks argues against artifacts: if results were driven by feature conflation, higher-fidelity SAEs would produce uneven rather than uniformly better results. Moreover, even K=64 SAEs outperform neuron and PCA probes at matched sparsity, confirming the SAE basis captures routing-relevant structure beyond standard bases.
>
> **Applicability to Expert Choice Routing (Q4)**
>
> Both SSH and RouterInterp apply to Expert Choice (EC) routing without significant modifications. **Argument 1**: regardless of routing direction, the assignment is optimized for the same loss—the incentive to minimize interference by grouping dissimilar micro-domains remains. **Argument 2**: under EC, each expert must fill a fixed capacity every pass, forcing generalism across micro-domains. RouterInterp's pipeline is agnostic to the assignment mechanism—ρ-usefulness is defined over empirical routing decisions regardless of how they arise.
>
> **Feature Sharing Across Experts (Q5)**
>
> We have added an  appendix with analysis of feature sharing. For gpt-oss-20b, only **18.4%** of each expert's top-10 ρ-useful features are unique. Under top-k routing, a feature firing on a token is scored high for all co-selected experts, revealing co-selection structure. However, shared features fire largely independently (see **W3, Reviewer 7XDd**) rather than as specific combinations.
>
>
> **Model Coverage (W4)**
>
> We agree with this and have added a discussion in the Limitations section. That said, many frontier MoEs share the same regime—decoder-only transformers with sparse top-k routing—including Mixtral, Qwen3.5 (256 experts, top-8), and Kimi K2 (384 experts, top-8). Our setting aligns with this common class while broader validation remains for future work.
>
> **Incomplete Baseline Comparisons (W5)**
>
> Prior MoE interpretation works (ST-MoE, Mixtral, OpenMoE, MoE Lens, Goodfire) all rely on token-level statistics, which our n-gram baseline formalises. To close the gap, we added sparse linear probes on neuron- and PCA-bases for routing prediction (**W2, Reviewer Z96h**) and expert-activation AutoInterp without SAE decomposition for explanations (**Q1, Reviewer 7XDd**). We updated the abstract claim accordingly. We already include a geometric-style comparison in the Appendix ("Feature Selection Method and Set Size") evaluating an alternative ranking features by cosine similarity between router weight vectors and SAE decoder directions against our ρ-usefulness rule.
>
> ---
>
> We hope we have addressed all concerns and welcome any additional questions.

---

> > ### Author Rebuttal · Reviewer_L2xV · 2026-04-02
> >
> > I view the evidence for SSH as promising but not fully conclusive because the paper gives strong evidence that is consistent with SSH, but it does not fully rule out other explanations. The added analyses make the SSH story more credible. For that reason, I still see this part of the paper as promising but not yet fully convincing. The results show patterns that are consistent with SSH, but they do not yet fully rule out alternative interpretations or establish SSH as the unique scientific explanation of expert specialization. Overall, I found this to be an interesting paper, and I am keeping my score unchanged.

---

> > > ### Author Response · Authors · 2026-04-05
> > >
> > > We thank reviewer L2xV for their engagement and for recognising the SSH story as "promising" and "more credible" with the added analyses.
> > >
> > > With additional time, we were also able to address your question on causal evidence for SSH.
> > >
> > > **Causal evidence (Q2, W2):** Following previous work on SAE feature steering (Templeton et al., 2024; Arad et al., 2025), we performed a small-scale causal intervention experiment. For each of 8 experts in OLMoE (layer 10), we selected 5 top ρ-useful features and 5 random SAE features (matched by median ρ score) as controls. We then ran two interventions:
> > >
> > > - **Steering:** On tokens where the target expert was *not* in the router top-k during the clean forward pass, we added the SAE feature direction (scaled by α × max activation) to the residual stream at the SAE hookpoint and re-ran the forward pass. We measured whether this caused the expert to enter the top-k. (α=1)
> > > - **Ablation:** On tokens where the target expert *was* in the router top-k, we subtracted the feature direction, measuring whether this knocked the expert out of the top-k.(α=-1)
> > >
> > > Our primary metric is the change in fraction of tokens which activate the expert (i.e. the expert appears in the router top-k), reported in percentage points (pp). Averaged over per-expert means, ρ-useful features increase expert activation by **+5.4 pp** under steering (per-expert means range from +0.5 to +14.0 pp) and decrease it by **−26.9 pp** under ablation (per-expert means range from −13.3 to −51.2 pp). The mean ρ-useful feature effect exceeds the mean random feature effect for 7/8 experts under steering and 6/8 under ablation (~2M tokens). This provides initial causal evidence that ρ-useful SAE features are not merely correlated with routing decisions but can directly influence them. We note this is a small experiment on a single layer of one model and leave scaling it to future work.
> > >
> > > ---
> > >
> > > Given that the human validation you identified as sufficient for raising your score has been provided, and now with causal analysis, could you let us know what specifically would be required for you to consider updating your score? We would love to discuss any remaining points.

---

### Official Review · Reviewer_Z96h · 2026-03-13

**Soundness:** 2
**Presentation:** 2
**Significance:** 2
**Originality:** 2
**Overall Recommendation:** 3
**Confidence:** 4

**Summary:**

This paper aims to predict and understand the mixture of experts routing in pre-trained LLMs. It raises two possible hypotheses: domain specialisation vs superposed specialisation, with the latter being analogous to neuron superposition. It proposes a method called RouterInterp, which trains sparse autoencoders on the LLM activations and uses them to predict the activated experts along with natural language explanations. Results demonstrate that this outperforms router prediction from n-gram statistics.

**Compliance With Llm Reviewing Policy:**

Affirmed.

**Final Justification:**

Thanks to the authors for engaging with the rebuttal. My concerns have been partially addressed and my current assessment is borderline. My concern is that the central claim about DSH vs SSH is not adequately supported, I would encourage the authors to revise, expanding on the preliminary analysis shared during this discussion period.

**Key Questions For Authors:**

See strengths and weaknesses. Other questions:

How is the performance of the SAE? RouterInterp depends on well-trained SAE concepts, this should be reported (reconstruction loss, variance explained, etc).

How consistent are the routing predictions across different layers?

Is the linear classifier trained on SAE features simply thresholded to predict E_pred? How calibrated are the SAE predictions (i.e. how well do the predicted probabilities assigned to each expert align with the actual MoE probabilities).

In Eq 6, is this an average over each token of the test set?

**Limitations:**

yes

**Strengths And Weaknesses:**

**Strengths**

The paper challenges a popular heuristic understanding of mixture-of-experts, which they term the Domain Specialisation Hypothesis. It is an interesting and timely question whether this intuition actually holds up. The contrast to the alternative Superposed Specialisation Hypothesis is well-motivated.

Novel application of SAEs, and straightforward for the community to grasp.

RouterInterp strongly outperforms an n-gram baseline.

**Weaknesses**

Evaluation of RouterInterp is incomplete:
* n-gram baseline is a weak baseline. RouterInterp should be compared with previous works on MoE interpretability, some of which are mentioned in A.2
* Because the only comparison is with n-grams, we don’t know whether RouterInterp’s gains are due to SAE features or simply having a richer representation (accessing continuous neural representations from the residual stream) than n-grams. A baseline could be a probe on the activations.

I am not convinced that the results support SSH over DSH. The paper shows that SAE-based classifier predicts routing better than n-g ram baseline, and also that natural language explanations from RouterInterp predict routing better than explanations from n-gram baseline. The question of accuracy is separate from whether the features are semantically disjoint, and the work lacks either a DSH-motivated baseline or more systematic analysis of whether the features activated by RouterInterp are disjoint or similar.

Training SAE is computationally expensive.

Minor: The paper can present more clearly the choices of k and m and the number of active experts in each model. How are k and m chosen? Also note the overloaded k (“k-sparse” SAE as well as later the number of experts in Eq 6).

---

> ### Author Rebuttal · Authors · 2026-03-31
>
> We thank Reviewer Z96h for their constructive feedback. Please find our detailed responses below.
>
> **Comparison with Prior MoE Interpretability Methods (Weakness 1)**
>
> We appreciate the suggestion. To our knowledge, prior works in Section A.2 do not provide alternative post-hoc methods as direct baselines. Works interpreting trained MoEs (ST-MoE, Mixtral, OpenMoE, MoE Lens, Goodfire) rely on token-level statistics, which our n-gram baseline formalizes. Remaining works train new architectures (MoE-X, not applicable post-hoc), study toy settings, or address different questions. We welcome specific suggestions on baselines from the reviewer. We also provide stronger baselines below.
>
> **Distinguishing SAE Features from Richer Representations (Weakness 2)**
>
> We added two baselines: **sparse linear probes on the neuron basis** (top-$k$ activating neurons from the residual stream) and **on PCA components** (top-$k$ after projecting onto leading principal components), both at matched sparsity. We also updated the metric to **macro-F1** (Section 5). Following **Reviewer L2xV's Q3**, we reran with a better SAE ($k{=}128$). Updated **Table 2** on gpt-oss-20b:
>
> | Method | Layer 4 | Layer 8 | Layer 12 | Layer 16 | Layer 20 | Avg |
> | :--- | :---: | :---: | :---: | :---: | :---: | :---: |
> | SAE ($k=128$) | **0.575** | 0.708 | **0.729** | 0.421 | **0.796** | **0.646** |
> | SAE ($k=64$) | 0.552 | **0.725** | 0.685 | 0.297 | 0.773 | 0.606 |
> | PCA Probe ($k=128$) | 0.544 | 0.623 | 0.536 | 0.494 | 0.366 | 0.513 |
> | PCA Probe ($k=64$) | 0.517 | 0.603 | 0.523 | 0.482 | 0.384 | 0.502 |
> | Neuron Probe ($k=128$) | 0.519 | 0.638 | 0.586 | **0.559** | 0.630 | 0.586 |
> | Neuron Probe ($k=64$) | 0.476 | 0.603 | 0.509 | 0.524 | 0.570 | 0.536 |
>
>
> SAE features achieve the highest average macro-F1 (0.646 with $k{=}128$, 0.606 with $k{=}64$), outperforming both neuron probes (0.586) and PCA probes (0.513) at matched sparsity.
>
> **Evidence for Superposed Specialization Hypothesis (Weakness 3)**
>
> We added **Section 6.3: Evidence for Superposed Specialisation** with systematic analysis testing whether expert features are disjoint. Details in **Reviewer 7XDd's (W3, Q2)**.
>
>
> **Computational Cost of SAE Training (Weakness 4)**
>
> Relative to LM pretraining, SAE training is minimal: Gao et al. (2025) estimate $10^{-5}$ to $10^{-9}$ of pretraining FLOPs. Our OLMoE SAEs trained in $\sim$2 hours on a single A100 (<$2). Importantly, RouterInterp does not require training new SAEs—open SAEs exist for many models (Gemma Scope, Llama Scope, GPT-2, Llama 3); for gpt-oss-20b we exclusively used publicly available pretrained SAEs. We added an appendix detailing computational costs.
>
> **Clarification of Notation and Hyperparameters (Weakness 5)**
>
> Thank you for raising this. We revised notation: $k$ = MoE routing (experts per token), $s$ = SAE latent sparsity (nonzero latents after Top-K), $m$ = SAE features per expert classifier. For OLMoE, $s=32$.
>
> **SAE Quality Metrics & Layer Consistency (Questions 1 & 2)**
>
> We added an appendix reporting FVU and dead features. SAE metrics for gpt-oss-20b (BatchTopK; 131,072 features, 45× expansion):
>
> | Layer | FVU (K=64) | FVU (K=128) | Dead % (K=64) | Dead % (K=128) |
> |-------|------------|-------------|---------------|-----------------|
> | 4     | 0.050      | 0.037       | 0.04          | 0.90            |
> | 8     | 0.101      | 0.077       | 0.02          | 0.29            |
> | 12    | 0.172      | 0.137       | 0.00          | 0.01            |
> | 16    | 0.232      | 0.191       | 0.00          | 0.00            |
> | 20    | 0.146      | 0.117       | 0.02          | 0.00            |
>
> Macro-F1 (see W2) is mostly consistent except **Layer 4** (early layers lack meaningful semantics) and **Layer 16**, where FVU is highest, directly reducing routing prediction. The $k{=}128$ SAE improves average F1 from 0.606 to 0.646, confirming SAE quality as a bottleneck. Full ablation in **Q3 of Reviewer L2xV**.
>
> **Classifier Training and Calibration (Question 3)**
>
> Yes, classifiers are thresholded at 0.5. We computed Spearman $\rho$ and MSE between predicted ($s{=}128$) and true router probabilities on gpt-oss-20b (added to appendix):
>
> | Layer | Spearman $\rho$ | MSE | F1 |
> | :---: | :---: | :---: | :---: |
> | 4 | 0.258 | 0.0278 | 0.575 |
> | 8 | 0.809 | 0.029 | 0.708 |
> | 12 | 0.741 | 0.030 | 0.729 |
> | 16 | 0.413 | 0.0285 | 0.421 |
> | 20 | 0.817 | 0.030 | 0.796 |
>
>
> Despite binary training, predicted probabilities align well with true distributions. Calibration strongly correlates with macro-F1—both are strong in middle/late layers and weaker at layers 4 and 16.
>
> **Averaging in Equation 6 (Question 4)**
>
> Eq 6 originally averaged over tokens (line 321). For clarity, we now use expert-level macro-F1 (see **W2**).
>
> ***
>
> Given the added neuron/PCA baselines (W2), SAE quality metrics (Q1/Q2), calibration analysis (Q3), and direct SSH tests (W3), we would appreciate if the reviewer considers raising their score. We welcome further discussion.

---

> > ### Author Rebuttal · Reviewer_Z96h · 2026-04-04
> >
> > Thanks for the clarifications and additional results, which I have found useful. I have updated my score.

---

> > > ### Author Response · Authors · 2026-04-05
> > >
> > > We would like to thank Reviewer Z96h for their time and for updating their score. We appreciate the support for our paper!

---

### Official Review · Reviewer_g2yB · 2026-03-13

**Soundness:** 2
**Presentation:** 2
**Significance:** 2
**Originality:** 2
**Overall Recommendation:** 4
**Confidence:** 3

**Summary:**

This paper aims to tackle MoE routing interpretability and argues for the Superposed Specialisation Hypothesis instead of a single-domain specialisation view. The main idea is: rather than explaining an expert by one coherent domain, the paper explains routing as a combination of multiple SAE features. It uses SAE features to predict expert routing with linear classifiers, and builds RouterInterp by selecting top \rho-useful features, generating feature explanations, and aggregating them into expert-level explanations.

**Compliance With Llm Reviewing Policy:**

Affirmed.

**Key Questions For Authors:**

n/a

**Limitations:**

yes

**Strengths And Weaknesses:**

Strengths:
1. The results appear strong within the experimental setup, with clear gains over the unigram and bigram baselines in Table 1 to Table 3.
2. The paper is reasonably original in how it combines a specific interpretability hypothesis about MoE experts with SAE-based feature selection and explanation aggregation.
3. I see the paper as addressing a relevant and timely problem. MoE routing is important and still poorly understood, so a method that makes routing more interpretable could be useful.

Weaknesses
1. The baseline set is fairly limited and is centered mainly on token co-occurrence statistics (Section 4.3). This makes the empirical case convincing for “SAE-feature-based explanations outperform simple token-level baselines,” but less conclusive for broader claims about MoE routing interpretability or the validity of SSH.
2. My suggestion for presentation improvement is mainly about claim calibration. The paper would be clearer if it more explicitly distinguished between showing that feature-based explanations outperform simple token-based baselines and showing that SSH is the right scientific interpretation of MoE experts.
3. The paper’s strongest empirical evidence is still somewhat indirect relative to its strongest conceptual claim. Table 1 shows that SAE features predict routing better than unigram and bigram baselines, and Table 3 shows that the resulting explanations receive better explanation scores. However, this more directly supports the claim that SAE-based feature explanations are more predictive than token-level statistics, not necessarily that the Superposed Specialisation Hypothesis is uniquely established as the correct account of expert specialization. The gap between “better predictive explanatory features” and “validation of SSH” still feels somewhat large.

---

> ### Author Rebuttal · Authors · 2026-03-31
>
> We thank Reviewer g2yB for their thoughtful feedback. We agree that the problem is timely, and appreciate the positive remarks on our experimental results. Please find our detailed responses below.
>
>
> **Expanding the baseline set beyond token co-occurrence statistics (Weakness 1)**
>
>
> We appreciate the reviewer's observation that our original baseline set was centered primarily on token-level statistics. To strengthen the empirical case beyond outperforming token-level baselines, we have added several new baselines for both the routing prediction task (**Section 6.1**)  and for RouterInterp (**Section 6.2**).
>
>
> For the routing prediction task (**Table 2**), we introduced sparse linear probes trained on model activations (see details in our response to **Reviewer Z96h’s W2**). SAE features (avg F1 = 0.646) outperform both neuron probes (0.586) and PCA probes (0.513). This result shows that SAEs serve as a better representational tool for analysing routing than n-grams or neurons.
>
> For RouterInterp (**Table 3**), we added a new baseline, **Expert Activations AutoInterp** - we collected the samples that activate an expert and prompted an LLM to find patterns and generate per-expert explanations (see detailed description in our response to **Reviewer 7XDd’s Q1**). RouterInterp outperforms Expert Activations AutoInterp in 4 out of 5 layers. We hypothesize this performance gap arises because experts respond to a superposition of unrelated micro-domains, which we believe is an evidence for SSH. By first disentangling these domains via SAE features and then recombining their explanations, our approach captures expert behaviour more faithfully than trying to extract a single coherent pattern from activating examples.
>
>
> **Calibrating claims and distinguishing predictive utility from SSH validation (Weakness 2)**
>
> Thank you for this suggestion regarding presentation clarity. We have made several revisions to more explicitly distinguish between demonstrating that feature-based explanations outperform token-based baselines and establishing SSH as the correct scientific interpretation:
>
> 1. As mentioned in **[Reviewer 7XDd](https://openreview.net/forum?id=rDNCWfRd69&noteId=f4LWFFO1Ld)’s W3** discussion, we added a new results subsection: **6.3: Evidence for Superposed Specialisation**. That directly separates the expert prediction/RouterInterp results from the specialisation hypothesis testing.
>
> 2. Additionally, we calibrate some parts of the paper:
>
>     * **Lines 373-376** now read:
>     	 > While this establishes SAE features as better predictive units for routing compared to token-level statistics, it does not by itself distinguish between the DSH and SSH; we provide direct tests of SSH-specific predictions in Section 6.3.
>
>     * **Lines 92-99** (third contribution bullet point) now read:
>      	> We provide three lines of evidence to suggest that expert specialisation is best understood through the Superposed Specialisation Hypothesis: the idea that experts specialise in a disjoint rather than semantically coherent collection of features. (i)$\sim$SAE features spanning unrelated micro-domains predict routing far better than token-level statistics, (ii)$\sim$feature co-activation rates within experts match the independence predicted by SSH rather than the correlation predicted by DSH, and (iii)$\sim$per-expert routing entropy over The Pile subsets reveals no domain preference. Together, these results indicate that routing is mediated by a combination of features rather than any single coherent domain (Section 6).
>
>
> **Bridging the gap between predictive features and SSH validation (Weakness 3)**
>
> Thank you for raising this point about providing evidence for SSH beyond predictive performance. We believe that new **6.3: Evidence for Superposed Specialisation** section (described in **[Reviewer 7XDd](https://openreview.net/forum?id=rDNCWfRd69&noteId=f4LWFFO1Ld)’s W3**) narrows that gap. These two experiments do not depend on explanation quality — they test structural and statistical properties of the routing itself.
>
> ***
>
> We hope we have addressed all of the reviewer's concerns and are happy to answer any additional questions the reviewer might have.

---

> > ### Author Rebuttal · Reviewer_g2yB · 2026-04-03
> >
> > Thank you for the detailed rebuttal. My concerns have been resolved. I will the positive score.

---

> > > ### Author Response · Authors · 2026-04-05
> > >
> > > We thank reviewer g2yB for confirming that their concerns have been resolved.
> > > Since all three original weaknesses are now resolved, we wanted to ask: is there anything further we could address or clarify that would lead you to consider raising your score? We would greatly appreciate any guidance and remain available for further questions.

---

### Official Review · Reviewer_7XDd · 2026-03-13

**Soundness:** 3
**Presentation:** 4
**Significance:** 3
**Originality:** 3
**Overall Recommendation:** 4
**Confidence:** 4

**Summary:**

This paper studies expert specialisation in MoE models. The authors present the Superposed Specialisation Hypothesis (SSH), which proposes that each expert specialises in a disjoint union of many fine-grained micro-domains rather than a single coherent domain. The paper provides a helpful theoretical discussion of how this specialisation pattern may naturally arise from training incentives, specifically load balancing and interference minimisation under superposition. The authors then build RouterInterp, a method that uses SAE features predictive of routing behaviour to generate natural language explanations of when each expert activates. They evaluate on gpt-oss-20b and OLMoE-1B-7B, showing that SAE-based routing prediction and explanation substantially outperform unigram and bigram baselines. They verify that RouterInterp's explanations are meaningful using an auto-interpretability scoring framework, where a language model predicts expert activation given only the natural language explanation.

**Compliance With Llm Reviewing Policy:**

Affirmed.

**Final Justification:**

I'm maintaining my weak accept, though I strongly considered bumping it to accept. None of my original questions were real blockers, and I appreciate that the authors ran some interesting additional analysis and and a new autointerp baseline. Ultimately I decided against raising my score because the study is limited to interpreting a few trained MoE models — which is perfectly reasonable for a paper, but also limits my excitement.

**Key Questions For Authors:**

1. A natural baseline would be to give an LLM randomly sampled text examples annotated with expert activations directly, without the SAE decomposition step. The n-gram baselines are clearly insufficient, but it is unclear whether the improvement comes from the SAE features specifically or from providing richer context than token counts. Such a baseline would help isolate the contribution of the SAE decomposition to explanation quality, and if done well would strengthen my assessment of the paper.

2. A more systematic qualitative analysis of the feature groups associated with each expert would be valuable. The paper includes one worked example (Expert 8), but a broader analysis examining whether the top features per expert consistently span unrelated micro-domains across many experts would provide much stronger evidence for the SSH. This would significantly strengthen my confidence in the connection between the empirical results and the hypothesis.

**Limitations:**

yes

**Strengths And Weaknesses:**

**Strengths**

* Expert specialisation in MoE models is an underexplored and scientifically interesting problem. This paper makes a meaningful contribution to an area that has received surprisingly little rigorous attention.
* The theoretical motivation for the two alternative specialisation hypotheses (DSH vs SSH) is well-articulated, with clear reasoning about how load balancing and interference minimisation incentivise superposed rather than domain-coherent specialisation.
* SAE features outperform unigram and bigram baselines both when used directly as inputs to a logistic classifier for routing prediction and when used as inputs to an auto-interp pipeline for generating natural language explanations of expert activation patterns.

**Weaknesses**

* The natural language explanations significantly underperform the SAE-based classifier (0.62 F1 vs 0.81 recall), suggesting that a substantial portion of routing behaviour is not captured by the human-understandable descriptions.
* The theoretical arguments for the SSH are compelling but remain purely conceptual. Even a toy-scale experiment — e.g., training a small MoE and observing how specialisation patterns emerge under different load-balancing regimes — would substantially strengthen the theoretical claims.
* The extent to which the SAE results support the SSH requires deeper analysis. A more qualitative investigation of the types of SAE feature groups associated with each expert — examining whether the features truly span unrelated micro-domains as the SSH predicts — would be exciting and would make the connection between the empirical results and the hypothesis more convincing.

---

> ### Author Rebuttal · Authors · 2026-03-31
>
> We would like to thank Reviewer 7XDd for their thoughtful feedback. Please find our detailed responses below.
>
> **Comparison between explanation scores and SAE-based classifier recall (Weakness 1)**
>
> We appreciate this observation; however, the two scores evaluate structurally different tasks. First, explanation scoring uses hard negatives (Section 4.4) while routing prediction uses naturally distributed tokens. Second, the two operate on vastly different information budgets. The SAE classifier is trained on all k=64 active latents per token over 1M tokens, observing $\sim$100,289 unique features out of 131,072 total (Section 5). RouterInterp uses only the top-10 $\rho$-useful features per expert, yielding $\sim$162 unique features across all 32 experts—$\sim$600× fewer. We updated Section 5 to make this contrast explicit.
>
> This reduction is expected: RouterInterp aims to find a *minimal, human-understandable* description of each expert's routing behaviour. Given this conciseness–accuracy trade-off, we do not believe the gap indicates routing behaviour escapes human-understandable descriptions, but acknowledge explanation scores are not maximal and future work remains.
>
> **Empirical validation of the SSH through toy-scale experiments (Weakness 2)**
>
> We agree that toy-model validation would be valuable; this work focused on interpreting trained MoE models at scale. We added a Future Work discussion on toy experiments detailing the phase boundary between specialisation patterns under different load-balancing regimes.
>
> **Qualitative investigation of SAE feature groups supporting the SSH (Weakness 3, Question 2)**
>
> We agree that qualitative investigation would strengthen our results. We have added a new subsection **6.3: Evidence for Superposed Specialisation** with two experiments testing whether expert-predictive features represent disjoint micro-domains:
>
> **Feature co-activation**: For each expert, we measure pairwise co-activation rates among its top-20 $\rho$-useful features over ${\sim}$10M tokens. We normalise observed co-activation by the independence baseline $\bar{p}^2$, yielding a ratio of 1.0 when features fire independently. Across 128 experts in gpt-oss-20b, observed co-activation closely follows the $\bar{p}^2$ independence curve rather than the $\bar{p}$ perfect-correlation line, supporting disjoint micro-domains consistent with the SSH.
>
> **The Pile routing**: The Pile consists of diverse subsets (ArXiv, PubMed, GitHub, etc.). Following analysis in Jiang et.al (2024), for each expert E_i, we computed the proportion of tokens from each subset that are routed to E_i, yielding a per-expert distribution over subsets. We quantify domain specialisation via normalised entropy: the Shannon entropy of each expert's subset distribution divided by $\log N_\text{subsets}$, so that **1.0**-  perfectly uniform distribution, **0.8** - corpus proportional. With a **violin plot** of distribution of normalized entropy per layer we show that expert entropies cluster near **0.8** at every layer, indicating that routing reflects corpus composition rather than domain preference.
>
>
> **LLM-based baseline without SAE decomposition (Question 1)**
>
> We appreciate the suggestion and we've implemented this new baseline. We renamed Section 4.3 to "Baselines" to describe both n-gram baselines and a new LLM-based baseline, **Expert Activations AutoInterp**:
>
> >To isolate the contribution of the SAE decomposition step, we compare against a baseline that applies the standard AutoInterp pipeline [Paulo et.al.] directly to expert activations without SAE decomposition. For each expert $E_i$, we collect text samples that activate the expert and prompt a language model to identify patterns and generate a natural language explanation. This baseline receives richer context than n-gram statistics (full text passages rather than token counts) but lacks the feature-level decomposition provided by SAEs. We report prompts used for the AutoInterp pipeline in Appendix E.
>
> Updated results in **Table 3**:
>
> |                               | Layer 4 | Layer 8 | Layer 12 | Layer 16 | Layer 20
> |-------------------------------|---------|---------|----------|----------|----------|
> | RouterInterp (SAE with k=128)                  | **0.59**     | **0.58**   | **0.59**     | 0.56     | **0.61**      |
> | RouterInterp (SAE with k=64)                  |   0.54  | 0.58   |  0.53 | 0.53 | 0.60  |
> | Expert Activations AutoInterp |     0.57    |      0.56   |       0.51   |    **0.63**      |     0.56     |
> | Bigram AutoInterp             | 0.23    | 0.47    | 0.42     | 0.35     | 0.40     |
>
> RouterInterp achieves the highest explanation score on 4 out of 5 layers, outperforming both baselines. RouterInterp (SAE k=128) now also reflects discussion from **Reviewer L2xV's Q1** regarding SAE variants.
>
>
> ***
>
> We hope we have addressed all of the reviewer's concerns and are happy to answer any additional questions the reviewer might have.

---

> > ### Author Rebuttal · Reviewer_7XDd · 2026-04-04
> >
> > **Acknowledgement:** (a) Fully resolved - My concerns have been adequately addressed. If you select this option, please consider adjusting your score accordingly.
> >
> > **Reasons:**
> >
> > Thank you for the detailed rebuttal.
> >
> >
> > **Q1:** Cool to see that both the LLM-based baseline and the SAE quality ablation independently add to the results.
> >
> >
> > **W1:** Resolved. The clarification on the information budget difference between the SAE classifier and RouterInterp explanations is helpful.
> >
> > **W2:** Resolved. I agree scoping this project to focus on interpreting MoEs at scale is reasonable
> >
> > **W3:** Mostly resolved. I'm excited to see the additional analysis. Upon rereading my weakness, I realize my ask wasn't fully clear. I think what I was more curious about is whether there is qualitative evidence that, despite being fragmented, there is still meaningful structure within the micro-domains — for example, if the top-K SAE features naturally grouped into G < K meaningful clusters.
> >
> >
> > I will maintain my positive review.

---

> > > ### Author Response · Authors · 2026-04-05
> > >
> > > We thank reviewer 7XDd for their positive assessment of our work, and we glad to see most of the questions resolved.
> > >
> > > **Regarding W3 (SAE feature clustering):** To explore whether top-K SAE features group into G < K meaningful clusters, we conducted a new clustering experiment. We embedded each experts rho-useful feature's natural-language explanation using a sentence transformer (all-MiniLM-L6-v2) and clustered the resulting vectors with k-means globally across 138 unique features for layer 11, gpt-oss-20b. The optimal sweep yielded 49 global clusters. Each expert's top-10 ρ-useful features span 7–10 of these clusters (mean = 8.5, median = 8), so G ≈ 8.5 ≈ K. This confirms that each expert's top predictive features are semantically diverse rather than concentrated in one domain, consistent with the SSH. This also aligns with findings in Appendix C where explanation quality saturates at ~10 features. We added this analysis with a UMAP visualisation to the paper.
> > >
> > > ---
> > > We are grateful that you have maintained your positive review. We would be happy to hear if there is anything further that would be required for you to consider raising your score.

---

### Decision · Program_Chairs · 2026-04-30

**Decision:**

Accept (regular)

**Comment:**

After reading the reviewers’ thoughtful evaluations and the authors’ substantial responses, I find that this paper still has some remaining issues. For example, one reviewer remains concerned that the central claim regarding DSH vs SSH is not yet fully supported. However, given the authors’ preliminary additional analyses during the discussion phase, the fact that most concerns from the other three reviewers were adequately addressed and they all provided positive assessments, and the paper’s own strengths in tackling the novel problem of expert specialisation with a well motivated method and clearly positive empirical results, I recommend that this paper be accepted.